# *Taar1* gene variants have a causal role in methamphetamine intake and response and interact with *Oprm1*

Alexandra M Stafford[1†], Cheryl Reed[1†], Harue Baba[1], Nicole AR Walter[2], John RK Mootz[1], Robert W Williams[3], Kim A Neve[1,4], Lev M Fedorov[5], Aaron J Janowsky[1,4,6], Tamara J Phillips[1,4]*

[1]Department of Behavioral Neuroscience and Methamphetamine Abuse Research Center, Oregon Health & Science University, Portland, United States; [2]Division of Neuroscience, Oregon National Primate Research Center, Portland, United States; [3]Department of Genetics, Genomics and Informatics, University of Tennessee Health Sciences Center, Memphis, United States; [4]Veterans Affairs Portland Health Care System, Portland, United States; [5]Transgenic Mouse Models Shared Resource, Knight Cancer Institute, Oregon Health & Science University, Portland, United States; [6]Department of Psychiatry, Oregon Health & Science University, Portland, United States

**\*For correspondence:**
phillipt@ohsu.edu

[†]These authors contributed equally to this work

**Competing interests:** The authors declare that no competing interests exist.

**Abstract** We identified a locus on mouse chromosome 10 that accounts for 60% of the genetic variance in methamphetamine intake in mice selectively bred for high versus low methamphetamine consumption. We nominated the trace amine-associated receptor 1 gene, *Taar1*, as the strongest candidate and identified regulation of the mu-opioid receptor 1 gene, *Oprm1*, as another contributor. This study exploited CRISPR-Cas9 to test the causal role of *Taar1* in methamphetamine intake and a genetically-associated thermal response to methamphetamine. The methamphetamine-related traits were rescued, converting them to levels found in methamphetamine-avoiding animals. We used a family of recombinant inbred mouse strains for interval mapping and to examine independent and epistatic effects of *Taar1* and *Oprm1*. Both methamphetamine intake and the thermal response mapped to *Taar1* and the independent effect of *Taar1* was dependent on genotype at *Oprm1*. Our findings encourage investigation of the contribution of *Taar1* and *Oprm1* variants to human methamphetamine addiction.
DOI: https://doi.org/10.7554/eLife.46472.001

## Introduction

Amphetamine-like stimulants, including methamphetamine (MA), remain the second most common class of illicit drugs used worldwide, behind cannabis-containing drugs (*United Nations Office on Drugs and Crime, 2016*). MA is the most widely abused psychostimulant (*Chomchai and Chomchai, 2015*), and the consequences of MA addiction pose major societal and health concerns (*National Institute on Drug Abuse, 2019*). MA-associated deaths in the United States were relatively stable between 2005 and 2013, but have since been on an increasing trajectory, rising from about 3600 individuals in 2013 to almost 11,000 in 2017 (*National Institute on Drug Abuse, 2018*). Discovering and understanding genetic factors that contribute to MA addiction risk, and uncovering the linked molecular and cellular processes, may help to curb this trend and improve interventions and long-term treatment success.

**eLife digest** People who misuse drugs often do so partly in response to the environment they find themselves in, and partly because of their genetics. The genetic component of someone's risk is influenced by many different genes, and most research has found that each gene has a small individual effect. A method called quantitative trait locus (QTL) analysis can help find parts of the genome that influence someone's risk of misusing drugs. In 2013, researchers found one region on chromosome 10 in mice has a particularly large influence on how much methamphetamine an individual mouse will ingest if the drug was available in one of its two water bottles. A gene called *Taar1* was particularly important in this region and another gene, called *Oprm1*, may also play a significant role.

When the *Taar1* gene is switched off, mice consume larger amounts of methamphetamine, have a heightened reward response from the drug, and are insensitive to the adverse effects – such as hypothermia. But whether *Taar1* directly caused these effects, and whether *Taar1* and *Oprm1* interact, had not yet been determined. If these genes played a causal role, they could be useful targets for treatment of methamphetamine-use disorder.

Stafford, Reed et al. – who include several of the researchers involved in the 2013 work – now report that when a particular variant of *Taar1* was present in mice they consumed large amounts of methamphetamine. The variant codes for a faulty version of a receptor protein. When this variant was replaced with a working version using gene editing, the mice consumed less methamphetamine and also became sensitive to hypothermia induced by the drug. This confirms that this gene does play a causal role in methamphetamine consumption and hypothermia. Next, Stafford, Reed et al. tested mice with different combinations of variants of *Oprm1* and *Taar1* to see how the genes interacted. The results showed that the effects of *Taar1* on both consumption of the drug and hypothermia depended on the *Oprm1* variant present.

The findings suggest that variants of these two genes in humans could influence an individual's risk of addiction to methamphetamine. It is possible that in future the disorder could be treated by drugs that modify the brain activity impacted by these receptors. But first, it will be important to find out if these genes play a similar role in humans as they do in mice.

DOI: https://doi.org/10.7554/eLife.46472.002

Whereas use of large human cohorts is effective at mapping loci and nominating candidate genes, precise experimental control over genetic composition, development, and drug exposure history is practical only in animal models. Evidence that mouse models continue to contribute critical information to our understanding of pathological conditions was recently reviewed (*Nadeau and Auwerx, 2019*). Selected lines of rodents have been derived to model genetic risk factors for drug use. These include, particularly, those bred for various drug intake phenotypes, such as our mice bred for different amounts of voluntary MA intake (*Hitzemann et al., 2019*; *Shabani et al., 2011*; *Wheeler et al., 2009*). In previous studies, we demonstrated a strong genetic contribution to level of voluntary MA consumption using multiple replicate sets of mice that we bidirectionally, selectively bred for MA high drinking (MAHDR) and MA low drinking (MALDR), which for simplicity are also referred to here as the 'high' and 'low' lines. Multiple lines of evidence support the involvement of the trace amine-associated receptor 1 gene, *Taar1*, in differences in MA intake, including: quantitative trait locus (QTL) mapping in the high and low lines (*Belknap et al., 2013*); the direct agonist activity of MA at the trace amine-associated receptor 1 (TAAR1) (*Bunzow et al., 2001*; *Wolinsky et al., 2007*); the presence of a single nucleotide polymorphism (SNP) in *Taar1* (*Taar1^{m1J}*) that renders the receptor it expresses (TAAR1) non-functional in response to MA and to other direct agonists (*Harkness et al., 2015*; *Shi et al., 2016*); fixation of the *Taar1^{m1J}* allele in high line mice of all five replicates, and retention of the wildtype, *Taar1^+* allele in all low line replicates (*Harkness et al., 2015*; *Reed et al., 2018*); greater level of MA intake in *Taar1* knockout mice compared to wildtype mice (*Harkness et al., 2015*); strong association of *Taar1* genotype with MA intake in multiple genetic models (*Reed et al., 2018*); and strong association of *Taar1* genotype with additional MA-related traits that correspond with level of MA intake, including level of sensitivity to

MA-induced hypothermia and conditioned taste aversion (*Harkness et al., 2015*; *Reed et al., 2018*).

In the current studies, to demonstrate a causal role for *Taar1*, we used CRISPR-Cas9 technology to replace the *Taar1$^{m1J}$* allele found in high line mice with the *Taar1$^+$* allele, creating a knock-in with functional TAAR1. These mice were then tested for MA consumption in comparison to an unaltered high line control, with the prediction that knock-in of *Taar1$^+$* would reduce MA intake. In addition, we tested knock-in and control mice for MA-induced change in body temperature to determine if this gene has a pleiotropic effect on this MA trait. We predicted that insertion of the *Taar1$^+$* allele would increase sensitivity to MA-induced hypothermia in the knock-in mice.

Our existing data indicate that *Taar1* genotype has a major impact on MA consumption, and gene mapping results demonstrate that a chromosome 10 QTL, at the location of *Taar1*, accounts for as much as 60% of the genetic variance in risk for MA intake (*Belknap et al., 2013*). Other undefined variants must account for the remaining genetic variance. We previously examined the μ-opioid receptor 1 gene, *Oprm1*, as a quantitative trait gene (QTG) residing at the proximal end of mouse chromosome 10, but it was eliminated as a risk factor for MA intake (*Eastwood et al., 2018*). Although *Oprm1* variants do not predict risk, *Oprm1* is involved in MA consumption via regulation by a top-ranked transcription factor network of genes that is differentially expressed between the MA drinking lines. We found that *Oprm1* and its product, the μ-opioid receptor (MOP-r), are more highly expressed in the prefrontal cortex of low line mice, relative to high line mice (*Belknap et al., 2013*; *Eastwood et al., 2018*). In addition, when MOP-r agonists were given to high line mice to test the hypothesis that increased MOP-r-regulated activity reduces MA intake, the acquisition of MA intake was attenuated, simulating the voluntary intake phenotype of the low line (*Eastwood et al., 2018*; *Eastwood and Phillips, 2014a*). High line mice possess the same mutant *Taar1$^{m1J/m1J}$* genotype as their progenitor DBA/2J (D2) strain and are more likely to also possess the *Oprm1* variant from the D2 strain, due to the relatively close proximity of *Taar1* and *Oprm1* on chromosome 10 (17 Mb apart). Due to this linkage, we have been unable to explore independent and interactive effects of the *Taar1* and *Oprm1* genotypes on MA intake or other MA-related traits in the MA drinking lines. Instead, we explored this in the present studies, using a family of related recombinant inbred mouse strains derived from a C57BL/6J (B6) x D2 inbred strain intercross (collectively the BXD strains) in which the linkage is disrupted. We examined the independent and interactive effects of these genes on MA intake and sensitivity to MA-induced change in body temperature, and performed QTL mapping for both traits using the BXD strain data.

## Results

### Knock-in of *Taar1$^+$* converts high MA intake to low MA intake

We predicted that MA intake would be reduced in high line mice in which the *Taar1$^{m1J}$* allele was removed and replaced with the *Taar1$^+$* allele. In the initial repeated measures analyses of variance (ANOVAs), there were no interactions involving sex for either mg/kg MA consumption or total volume consumed, so we performed additional analyses with data for the sexes combined. For MA consumption (*Figure 1a*; *Figure 1—source data 1*), there was a significant MA concentration x genotype interaction ($F$[1,36]=66.8, p<0.0001). Replacement of the *Taar1$^{m1J}$* allele with the *Taar1$^+$* allele converted high MA intake to low MA intake for both MA concentrations. Control mice, which are homozygous *Taar1$^{m1J}$*, consumed significantly more MA at the 40 mg/l concentration, compared to the 20 mg/l concentration, whereas low and comparable levels of MA consumption were found for both MA concentrations in knock-in mice. For total volume consumed from the water and MA tubes, during the time that MA was available (*Figure 1b*; *Figure 1—source data 1*), there was only a significant main effect of MA concentration ($F$[1,36]=21.4, p<0.0001). Total volume consumed was greater when mice were offered 40 mg/l MA, compared to 20 mg/l MA.

### Knock-in of *Taar1$^+$* restores a hypothermic response to MA in high line mice

Thermal response to MA is a genetically-correlated response to selection for level of MA intake; thus, low MA drinking line mice display MA-induced hypothermia, whereas high line mice do not (*Harkness et al., 2015*). We tested the hypothesis that *Taar1* genotype has a causal role. Knock-in

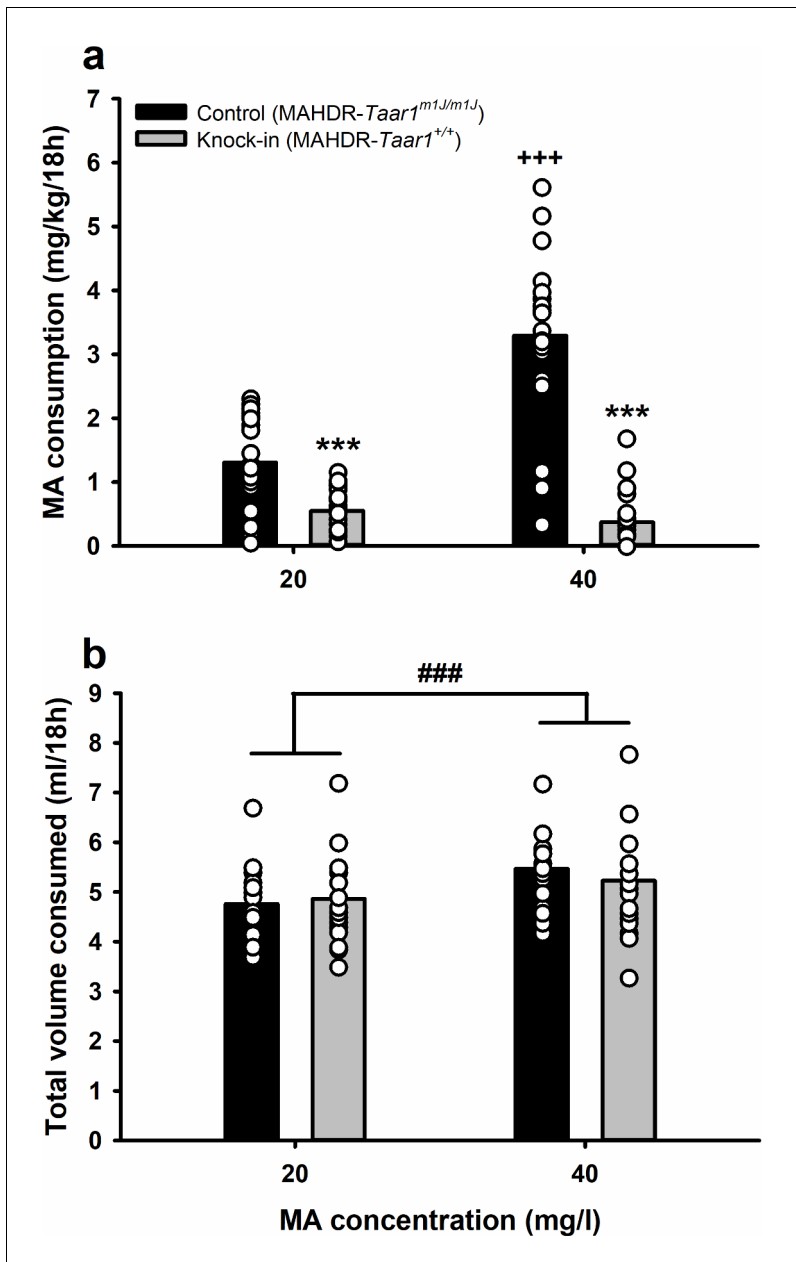

**Figure 1.** Knock-in of $Taar1^+$ converts high MA intake to low MA intake, but does not impact total volume of fluid consumed. (a) MA consumption and (b) total volume consumed, when MA was offered vs. water for 18h/day at a concentration of 20 or 40 mg/l for control and knock-in mice. Presented are means (represented by bars) and individual data points (represented by circles) for data collapsed on sex. n = 20/genotype tested in a single cohort. Repeated measures ANOVA followed by simple main effects analysis, ***p<0.001 vs. control (MAHDR-$Taar1^{m1J/m1J}$) at the same MA concentration; +++p < 0.001 vs. control (MAHDR-$Taar1^{m1J/m1J}$) at 20 mg/l MA. Repeated measures ANOVA, ###p<0.001 for the main effect of 20 mg/l vs. 40 mg/l MA. MA, methamphetamine; MAHDR, MA high drinking mice; $Taar1$, trace amine-associated receptor 1 gene; $Taar1^{+/+}$, homozygous for reference $Taar1^+$ allele; $Taar1^{m1J/m1J}$, homozygous for mutant $Taar1^{m1J}$ allele. The raw data represented in the graphs are available in *Figure 1—source data 1*.

DOI: https://doi.org/10.7554/eLife.46472.003

The following source data is available for figure 1:

**Source data 1.** MA consumption (mg/kg/18h) and total volume consumed (ml/18h) data for male and female, control MAHDR-$Taar1^{m1J/m1J}$ and knock-in MAHDR-$Taar1^{+/+}$ mice.

DOI: https://doi.org/10.7554/eLife.46472.004

of $Taar1^+$ in high line mice restored a hypothermic response to MA that did not occur in control mice (*Figure 2*; *Figure 2—source data 1*). In the initial repeated measures ANOVA there were significant independent interactions of sex with genotype ($F_{[1,90]}$=7.1, p=0.009), MA dose ($F_{[1,90]}$=4.9, p=0.03) and time ($F_{[4,360]}$=2.9, p=0.02). However, sex did not determine the pattern of response to saline vs. MA in each genotype across time. There was a significant 3-way genotype x MA dose x time interaction ($F_{[4,360]}$=10.2, p<0.0001) that did not interact with sex. We next examined the effects of genotype and time within each dose group. In the saline-treated mice (*Figure 2a*), there were significant main effects of time ($F_{[4,188]}$=35.9, p<0.0001) and genotype ($F_{[1,47]}$=4.7, p=0.04). Temperature declined by 0.3°C over time, regardless of genotype, and knock-in mice had 0.25°C higher body temperature than control mice, regardless of time. In the MA-treated mice (*Figure 2b*), the genotype x time interaction was significant ($F_{[4,188]}$=13.0, p<0.0001). Knock-in mice had a significant hypothermic response to MA at T30 compared to T0, whereas control mice were resistant to the hypothermic effects of MA, instead exhibiting significant MA-induced increases in body temperature at all post-MA treatment time points, compared to T0. Body temperatures were significantly higher in knock-in than control mice at baseline, by about 0.6°C, but significantly lower by about 1.2°C, at T30.

## MA consumption is impacted in BXD mice by a *Taar1* X *Oprm1* interaction

Initial analyses characterized 18 BXD strains for MA consumption (*Figure 3*; *Figure 3—source data 1*). There was a significant MA concentration x strain interaction ($F_{[17,215]}$=17.2, p<0.0001). MA intake differed among the strains for both the 20 (*Figure 3a*; p<0.0001) and 40 (*Figure 3b*; p<0.0001) mg/l MA concentrations, and all $Taar1^{m1J/m1J}$ strains, but not $Taar1^{+/+}$ strains, exhibited higher levels of MA intake from the 40 mg/l MA concentration than from the 20 mg/l concentration (p=0.0006 to<0.0001). For total volume consumed when mice were offered water vs. MA, there was a main effect of strain ($F_{[17,215]}$=27.7, p<0.0001) and a main effect of MA concentration ($F_{[1,215]}$=58.6, p<0.0001). Total volume consumed differed among the strains by as much as 4 ml (*Figure 3c*), but the strain distribution pattern for total volume did not correspond with the distributions for MA intake at either MA concentration. Total volume consumed was significantly less when water and 20 mg/l MA were available, compared to when water and 40 mg/l MA were available (mean ± SEM = 5.3±0.1 and 5.7 ± 0.1 for 20 and 40 mg/l, respectively).

We next examined the data from the BXD mice with regard to *Taar1* and *Oprm1* genotype to detect potential independent and epistatic effects on MA consumption. First, to identify potential independent effects, we calculated Pearson's correlations of each genotype with individual 20 and 40 mg/l MA intake amounts. MA intake was significantly associated with *Taar1* genotype (r = 0.74 and 0.81, *p*s <0.0001, for 20 and 40 mg/l MA, respectively), but not *Oprm1* genotype (r = 0.06 and 0.10, p=0.36 and 0.13, for 20 and 40 mg/l MA, respectively). Next, we considered main and interaction effects by repeated measures ANOVA with MA concentration, *Oprm1* genotype, *Taar1* genotype, and sex as factors. There were no significant interactions involving sex, so further analyses were performed with data for the sexes combined. Data are summarized in *Figure 4* (*Figure 4—source data 1*), with mice grouped according to their four possible *Taar1*/*Oprm1* genotypes (*Table 1*). For MA consumption (*Figure 4a*), there was a significant *Oprm1* genotype x *Taar1* genotype x MA concentration interaction ($F_{[1,225]}$=17.6, p<0.0001). For consumption of MA from the 20 mg/l MA concentration, main effects of the *Oprm1* ($F_{[1,229]}$=8.6, p=0.004) and *Taar1* ($F_{[1,229]}$=301.3, p<0.0001) genotypes indicated that both genes influenced MA intake, with greater MA intake in $Taar1^{m1J/m1J}$ and $Oprm1^{D2/D2}$ mice, compared to $Taar1^{+/+}$ and $Oprm1^{B6/B6}$ mice, respectively. However, there was no significant *Oprm1* genotype x *Taar1* genotype interaction. In contrast, for MA consumption from the 40 mg/l MA concentration, there was a significant *Oprm1* genotype x *Taar1* genotype interaction ($F_{[1,229]}$=13.9, p=0.0002). MA intake was significantly higher in $Taar1^{m1J/m1J}$ mice, compared to $Taar1^{+/+}$ mice, regardless of *Oprm1* genotype (*p*s <0.0001). However, $Taar1^{m1J/m1J}/Oprm1^{D2/D2}$ mice consumed significantly more MA than $Taar1^{m1J/m1J}/Oprm1^{B6/B6}$ mice. *Oprm1* genotype did not significantly impact MA intake in $Taar1^{+/+}$ mice.

For total volume consumed (*Figure 4b*), there were significant main effects of *Oprm1* genotype ($F_{[1,225]}$=24.8, p<0.0001), *Taar1* genotype ($F_{[1,225]}$=4.7, p=0.03), and MA concentration ($F_{[1,225]}$=75.7, p<0.0001), but no significant interactions. $Oprm1^{D2/D2}$ mice consumed more total volume

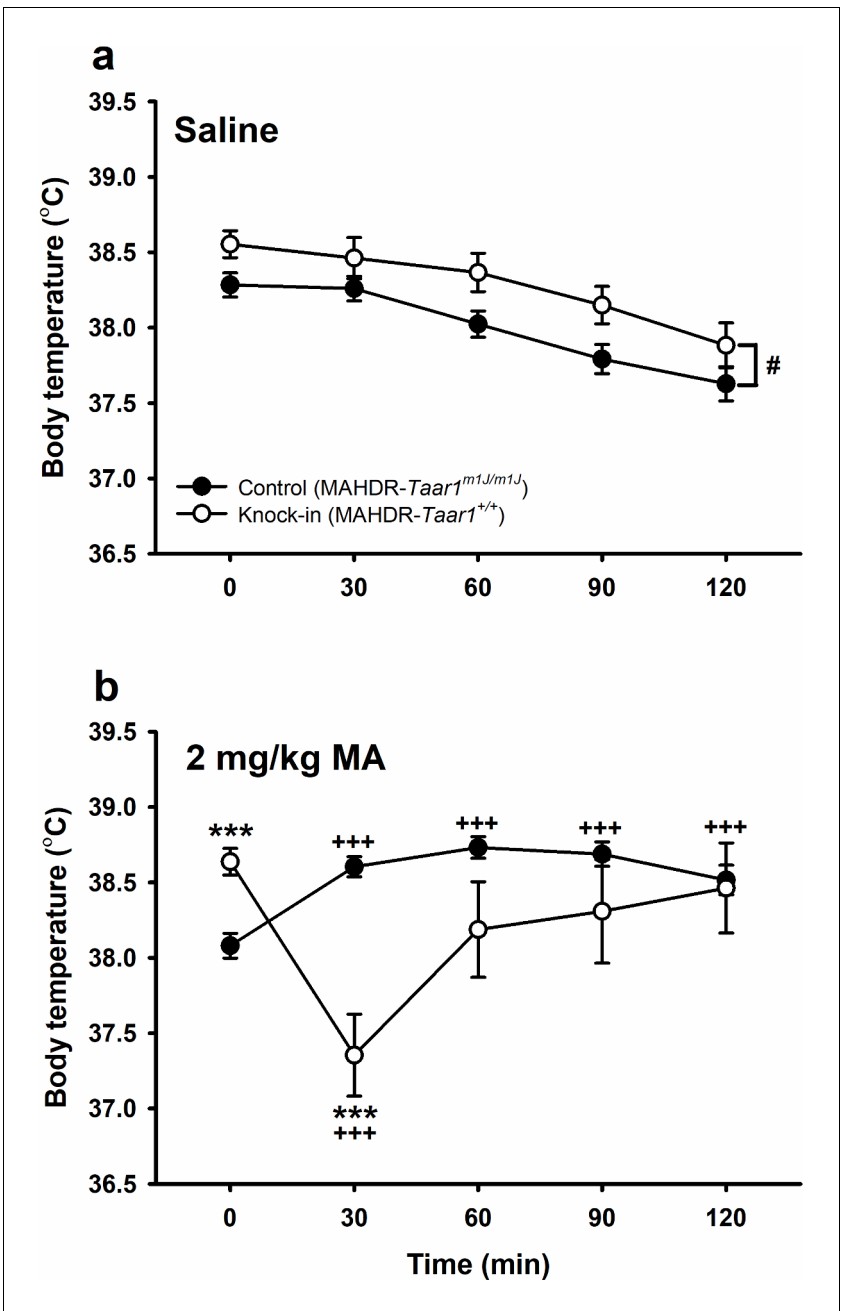

**Figure 2.** Knock-in of *Taar1*[+] restores a hypothermic response to MA. Body temperatures obtained immediately before (T0) or 30–120 min (T30-120) after (**a**) saline or (**b**) 2 mg/kg MA treatment for control and knock-in mice. Presented are means ± SEM for data collapsed on sex. n = 24–25/genotype/MA dose tested in 3 cohorts of 25–46 mice. Repeated measures ANOVA, #p<0.05 for the main effect of genotype. Repeated measures ANOVA followed by simple main effects analysis, ***p<0.001 for the effect of genotype at the indicated time point; Dunnett's post hoc test,+++*p* < 0.001 compared to T0. MA, methamphetamine; MAHDR, MA high drinking mice; *Taar1*, trace amine-associated receptor 1 gene; *Taar1*[+/+], homozygous for reference *Taar1*[+] allele; *Taar1*[m1J/m1J], homozygous for mutant *Taar1*[m1J] allele. The raw data represented in the graphs are available in ***Figure 2—source data 1***.

DOI: https://doi.org/10.7554/eLife.46472.005

The following source data is available for figure 2:

**Source data 1.** Core body temperature (˚C) data across time for saline and 2 mg/kg MA-treated male and female, control MAHDR-*Taar1*[m1J/m1J] and knock-in MAHDR-*Taar1*[+/+] mice.

DOI: https://doi.org/10.7554/eLife.46472.006

than $Oprm1^{B6/B6}$ mice, and $Taar1^{m1J/m1J}$ mice consumed more than $Taar1^{+/+}$ mice. In addition, total volume consumed was greater when mice were offered 40 mg/l MA, compared to 20 mg/l MA.

## MA-induced hypothermia is impacted in BXD mice by a *Taar1* X *Oprm1* interaction

We used a similar approach to analyze body temperature data that were collected after saline or MA treatment in 15 BXD strains (*Figure 5*; *Figure 5—source data 1*). For ease of viewing, the data in *Figure 5* are separated by saline/MA treatment and *Taar1* genotype for strain responses across time. A repeated measures ANOVA detected a significant strain x MA dose x time interaction ($F$ [56,1172]=10.0, p<0.0001). In saline-treated mice (*Figure 5a,b*), the BXD strains differed at each time point (*p*s = 0.04 to<0.0001), except T0, and body temperature differed across time in strains 161 ($Taar1^{+/+}/Oprm1^{D2/D2}$), 171, 194 (both $Taar1^{m1J/m1J}/Oprm1^{B6/B6}$), 186, 210 (both $Taar1^{m1J/m1J}/Oprm1^{D2/D2}$), and 218 ($Taar1^{+/+}/Oprm1^{B6/B6}$) (*p*s = 0.004 to<0.0001). In MA-treated mice (*Figure 5c,d*), the BXD strains differed at each time point (*p*s <0.0001), except T0, and body temperature differed across time in all strains (*p*s = 0.007 to<0.0001), except 160, 171, 194 (all $Taar1^{m1J/m1J}/Oprm1^{B6/B6}$), 178, 186, and 210 (all $Taar1^{m1J/m1J}/Oprm1^{D2/D2}$). The maximum mean drop in body temperature in saline-treated mice was 0.9°C (*Figure 5a,b*), whereas there was a maximum mean drop of 4°C in MA-treated $Taar1^{+/+}$ mice (*Figure 5c*), and a maximum increase of 1°C in MA-treated $Taar1^{m1J/m1J}$ mice (*Figure 5d*).

We next examined the BXD data with regard to *Taar1* and *Oprm1* genotype to detect potential independent and epistatic effects on the hypothermic effect of MA. Again, to identify potential independent effects, we calculated Pearson's correlations of each genotype with MA-induced change in body temperature from T0 to T30. There was a significant correlation of body temperature change with *Taar1* genotype ($r$ = 0.71, p<0.0001), but not with *Oprm1* genotype ($r$ = 0.11, p=0.16). Next, we considered main and interaction effects by repeated measures ANOVA with time, *Oprm1* genotype, *Taar1* genotype, sex and MA dose as factors. There were significant independent interactions of sex with MA dose ($F$[1,307]=8.0, p=0.006) and time ($F$[4,1228]=5.0, p=0.0007). However, sex did not play a significant role in the pattern of response of each genotype to each dose across time. There was a significant *Oprm1* genotype x *Taar1* genotype x MA dose x time interaction ($F$[4,1228] =8.0, p<0.0001) that did not interact with sex. We next examined the effects of genotype and time within each dose group (*Figure 6*; *Figure 6—source data 1*). In the saline-treated mice (*Figure 6a*), the *Oprm1* genotype x time interaction was significant ($F$[4,596]=2.6, p=0.04). Both $Oprm1^{B6/B6}$ and $Oprm1^{D2/D2}$ mice exhibited reductions in body temperatures at T60-120 (*p*s <0.001), compared to T0. However, these differences in body temperature were of no more than 0.3°C on average. In the MA-treated mice (*Figure 6b*), there was a significant 3-way interaction of *Oprm1* genotype x *Taar1* genotype x time ($F$[4,664]=11.4, p<0.0001). $Taar1^{+/+}/Oprm1^{D2/D2}$ mice had significantly lower body temperatures at all post-MA treatment time points, compared to $Taar1^{+/+}/Oprm1^{B6/B6}$ mice. $Taar1^{+/+}/Oprm1^{B6/B6}$ mice exhibited a significant hypothermic response to MA at T30-90, compared to T0, and recovered to baseline (T0) temperatures by T120, whereas $Taar1^{+/+}/Oprm1^{D2/D2}$ mice exhibited a significant hypothermic response to MA at T30-120, and remained below baseline for the duration of testing. However, the rate of increase after the initial decrease at T30 was similar for the two genotypes. Regardless of *Oprm1* genotype, $Taar1^{m1J/m1J}$ mice were resistant to the hypothermic effects of MA, exhibiting increases in body temperature at all post-MA treatment time points, compared to T0.

## MA consumption and body temperature response to MA map to a region of chromosome 10 at the location of *Taar1*

The QTL results for both traits are displayed in *Figure 7* (*Figure 7—source data 1*). A significant QTL in the same region of chromosome 10 emerged for both MA consumption (*Figure 7a*) and body temperature response to MA (*Figure 7b*). This QTL is at the location of the *Taar1* SNP (23.9 Mb). The correlations between the strain means and the *Taar1* SNP were r = 0.94, p<0.0001 for MA consumption and r = 0.82, p=0.02 for temperature response to MA. In addition, suggestive QTLs were detected on chromosomes 17 and 18 for MA effect on body temperature. There was a strong statistical trend for an increase in the chromosome 10 logarithm of the odds (LOD) score for MA intake (from 8.2 to 8.8; p=0.07), and a significant increase for the chromosome 10 LOD score for

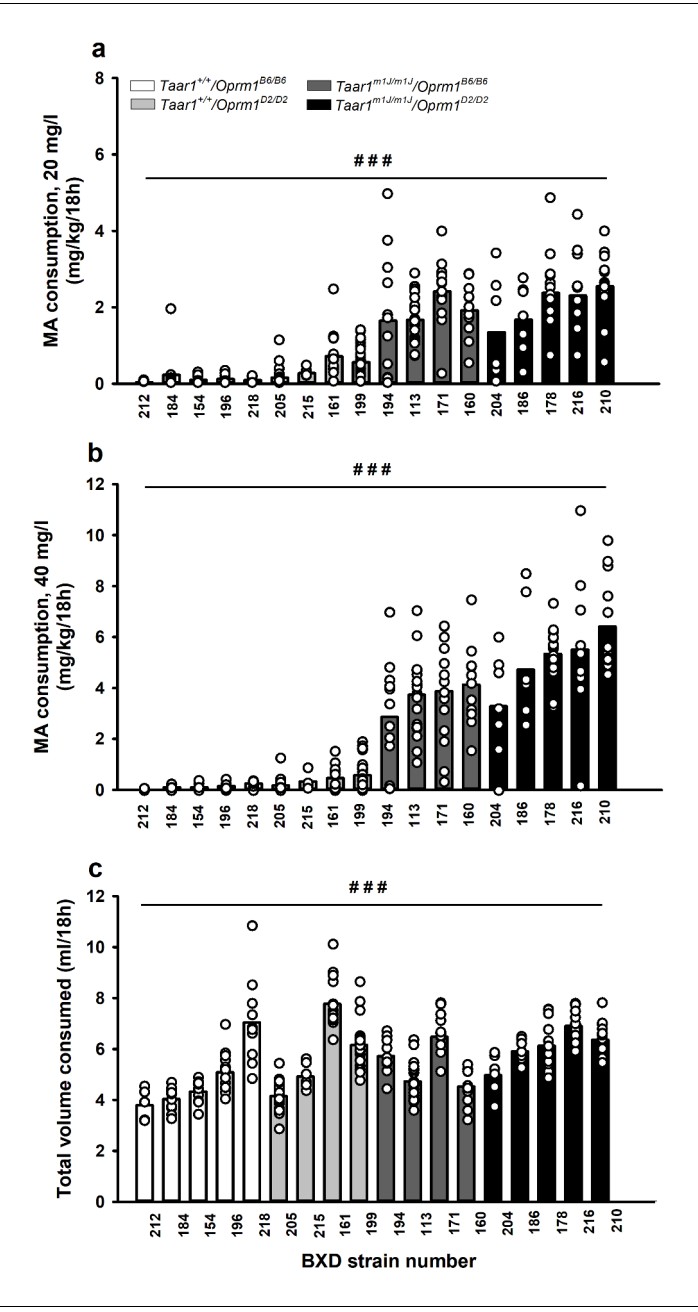

**Figure 3.** MA consumption and total volume consumed among BXD strains are strain-dependent. MA consumption when MA was offered vs. water for 18h/day at a concentration of (**a**) 20 mg/l or (**b**) 40 mg/l. (**c**) Total volume consumed from the water and MA tubes during the 18h period when MA was offered vs. water, collapsed on MA concentration, because there was no concentration-dependent effect. Presented are means (represented by bars) and individual data points (represented by circles) for data collapsed on sex. n = 5–28/strain tested in 4 cohorts of 27–97 mice. Repeated measures ANOVA, ###p<0.001 for the main effect of strain. BXD, C57BL/6J (B6) x DBA2/J (D2); MA, methamphetamine; *Oprm1*, mu opioid receptor gene; *Oprm1*$^{B6/B6}$, homozygous for B6 *Oprm1* allele; *Oprm1*$^{D2/D2}$, homozygous for D2 *Oprm1* allele; *Taar1*, trace amine-associated receptor 1 gene; *Taar1*$^{+/+}$, homozygous for reference *Taar1*$^{+}$ allele; *Taar1*$^{m1J/m1J}$, homozygous for mutant *Taar1*$^{m1J}$ allele. The raw data represented in the graphs are available in *Figure 3—source data 1*.

DOI: https://doi.org/10.7554/eLife.46472.007

The following source data is available for figure 3:

**Source data 1.** MA consumption (mg/kg/18h) and total volume consumed (ml/18h) data for male and female, BXD mice, according to *Taar1* and *Oprm1* genotype.

DOI: https://doi.org/10.7554/eLife.46472.008

temperature response (from 3.7 to 4.2; p=0.01), when results were compared for interval mapping and composite interval mapping. Composite mapping identified a suggestive QTL on chromosome four for MA intake that had not reached the suggestive significance threshold in initial mapping; however, the LOD change from 2.5 to 2.7 was not statistically significant (*Figure 7c*). The LOD scores for the suggestive QTLs on chromosomes 17 and 18 for temperature response were comparable for initial and composite mapping (*Figure 7d*).

## Discussion

Using CRISPR-Cas9 technology, we confirmed that *Taar1* is a QTG for MA consumption, with an apparent pleiotropic influence on sensitivity to MA-induced hypothermia. Data from the BXD mice indicate that *Taar1* and *Oprm1* interact in their effects on these traits. Furthermore, the BXD data confirmed the QTL on mouse chromosome 10 for MA consumption that we originally identified in the MA drinking lines (*Belknap et al., 2013*), and we mapped a QTL for MA-induced hypothermia to the same region in the BXD strains. *Taar1* resides at this location. Overall, the findings for *Taar1* are consistent with published results, including those demonstrating that *Taar1* knockout mice consume higher amounts of MA and are insensitive to MA-induced hypothermia, compared to their wildtype littermates (*Harkness et al., 2015*); that *Taar1* genotype is strongly correlated with MA consumption and MA-induced hypothermia in multiple genetic models on both homogeneous and heterogeneous backgrounds (*Reed et al., 2018*); and that TAAR1 agonists reduce MA self-administration (*Cotter et al., 2015*; *Jing and Li, 2015*). We conclude that TAAR1 activation has a causal role in protection against high levels of MA intake, and thus, has potential involvement in resistance to MA addiction.

### *Taar1* and MA consumption

We leveraged CRISPR-Cas9 technology to replace the mutant $Taar1^{m1J}$ allele with the common $Taar1^{+}$ allele in selectively bred high line mice. The patterns of MA consumption in mice with the different *Taar1* genotypes mimicked the patterns previously observed in the MA drinking lines (*Harkness et al., 2015*; *Shabani et al., 2011*; *Wheeler et al., 2009*). Thus, control mice escalated their MA consumption with increasing MA concentration, similar to previous findings in high line mice, whereas MA consumption was low and not concentration-dependent in the knock-in mice, similar to previous findings in the selectively bred low line mice.

Although oral is not the most common route of administration, some MA users do take MA orally, and when taken via this route, MA has a half-life similar to that for other routes of administration (*Cruickshank and Dyer, 2009*). Furthermore, oral consumption, like other routes of administration, can lead to dependence (*Galloway et al., 2010*). On average, mice with absent TAAR1 function consume about 6 mg/kg/18h from a 40 mg/l MA solution (*Harkness et al., 2015*; *Hitzemann et al., 2019*; *Reed et al., 2018*; *Shabani et al., 2011*; *Wheeler et al., 2009*). High line mice will consume an average of as much as 14 mg/kg/18h, when MA is offered as a 140 mg/l concentration (*Shabani et al., 2016*). Daily MA use in humans ranges from 300 to 800 mg/day on average (*Cho et al., 2001*; *Simon et al., 2002*), which translates to 3.9 to 10.4 mg/kg/day in a 77 kg individual. Whereas the half-life of MA is shorter in rodents compared to humans (*Cho et al., 2001*), relevant levels of MA are attained in our mouse model, with locomotor stimulant effects observed in the high line mice at consumed doses as low as 0.4 mg/kg in 1h (*Shabani et al., 2012a*).

In rats with functional TAAR1, TAAR1 agonists delivered systemically or to specific subregions of the mesocorticolimbic system, decreased MA seeking and cocaine seeking (*Liu et al., 2017*; *Pei et al., 2017*), suggesting that treatments that increase TAAR1 activity could be beneficial therapeutics. However, when TAAR1 is non-functional, receptor agonists are not an option, leaving us to consider downstream mechanisms of TAAR1 activation. Details of the mechanisms enlisted by TAAR1 activation remain elusive. Expressing TAAR1 using a heterologous expression system to identify its messenger system(s) has proved difficult, potentially due to the predominant intracellular localization of the receptor. Studying the function of TAAR1 is also complicated by the lack of good quality antibodies and selective antagonists (*Liu and Li, 2018*; *Rutigliano et al., 2017*). There is one available TAAR1 antagonist, EPPTB (*Bradaia et al., 2009*), but it has a high rate of clearance (*Rutigliano et al., 2017*; *Stalder et al., 2011*) and poor solubility, which limit in vivo use. Our

attempts to develop better antagonists with chemist collaborators have not yet met with success, and other investigators in the field have had a similar experience (*Lam et al., 2018*).

One strategy that could be considered for reducing MA consumption is to target mechanisms that are impacted by the absence of TAAR1 function. For example, TAAR1 modulates monoaminergic neurotransmission (*Espinoza et al., 2015a*; *Leo et al., 2014*; *Lindemann et al., 2008*; *Revel et al., 2011*; *Xie and Miller, 2008*), and recently has been implicated in glutamatergic neurotransmission (*Espinoza et al., 2018*; *Espinoza et al., 2015b*). High line mice are hyperglutamatergic at baseline, compared to low line mice, in both the nucleus accumbens (NAc) and medial prefrontal cortex (mPFC). They also exhibit a larger glutamate response to MA in the NAc, but not mPFC (*Lominac et al., 2016*; *Szumlinski et al., 2017*). In addition, compared to low line mice, high line mice have blunted dopamine levels in both the NAc and mPFC, and exhibit a heightened dopaminergic response to MA in the mPFC, but not NAc (*Lominac et al., 2014*). Whether these differences are related to *Taar1* genotype is unknown, but they suggest certain manipulations to be examined for their role in MA intake.

## Genetic basis for MA-induced thermal response

Our previous findings in the MA drinking lines demonstrated that the thermal response to MA is genetically-correlated with level of MA consumption. Thus, high line mice are resistant to the hypothermic effect of MA, whereas low line mice are highly sensitive to this effect of MA (*Harkness et al., 2015*). Here, we performed QTL analysis using data from the BXD strains and identified a significant QTL on chromosome 10 for the effect of MA on body temperature. This QTL is in the SNP-poor region of chromosome 10 (*Shi et al., 2016*), where we mapped the QTL for MA consumption in the MA drinking lines and BXD strains. QTL mapping was previously performed in BXD strains for the effects of 4, 8 and 16 mg/kg MA on body temperature recorded 48 min after administration. Although a QTL was mapped on chromosome 10 for the 4 mg/kg dose, it was considerably distal to the current QTL (*Grisel et al., 1997*), and not likely to be the same one for several reasons. First, the MA doses and time after administration were considerably different from those in our studies. But, more importantly, the *Grisel et al. (1997)* paper was published well before the *Taar1*$^{m1J}$ mutation arose in D2 mice (*Reed et al., 2018*); therefore, all of the BXD strains in that study shared the common *Taar1*$^{+/+}$ genotype. We have confirmed the *Taar1*$^{+/+}$ genotype of many of the BXD strains that were derived prior to when the mutation appeared (*Reed et al., 2018*; *Shi et al., 2016*).

## Relationship between MA consumption and body temperature

Sensitivity to MA-induced hypothermia corresponds with low levels of MA consumption in MAHDR-*Taar1*$^{+/+}$ knock-in mice, MALDR line mice, the wildtype littermates of *Taar1* knockout mice, and BXD strains with the *Taar1*$^{+/+}$ genotype. These data could indicate that *Taar1* has independent pleiotropic effects on the two traits. Another possibility is that the subjective experience of MA-induced hypothermia in mice with functional TAAR1 limits MA consumption, thereby playing a protective role. Not known is whether MA-induced hypothermia is subjectively unpleasant. However, hypothermia prolonged the associative period during which aversion could be conditioned in a conditioned taste aversion (CTA) procedure (*Misanin et al., 1998*; *Misanin et al., 2002*), and mice with functional TAAR1 form MA-induced CTA, whereas those without functional TAAR1 do not (*Harkness et al., 2015*; *Shabani et al., 2012b*; *Wheeler et al., 2009*). Low line and high line mice voluntarily consume similar amounts of MA on the first day that MA is offered, but low line mice then decrease consumption on the subsequent day, and remain at low intake levels from then on (*Eastwood et al., 2014*; *Shabani et al., 2012a*). It is possible that initial MA consumption induces hypothermia in low line mice, so that initial MA consumption is associated with this potentially unpleasant physiological effect, and reduces the desire to consume more MA. Additional research is needed to determine if MA drinking produces changes in body temperature, as occurs in response to an IP injection of MA.

Regardless of the chicken-egg question, the *Taar1*$^+$ allele replacement in high line mice produced MA intake levels like those found in low line mice, indicating that this gene has a major role in determining level of MA intake. Similarly, both the pattern and magnitude of hypothermic response to MA in the knock-in mice resembled those of low line mice. Low line mice had a maximal response of 1–2°C (depending upon replicate line) after treatment with 2 mg/kg MA (*Harkness et al., 2015*), and the response in the knock-in mice was also maximal at 30 min and of about 1.2°C. Therefore, *Taar1*

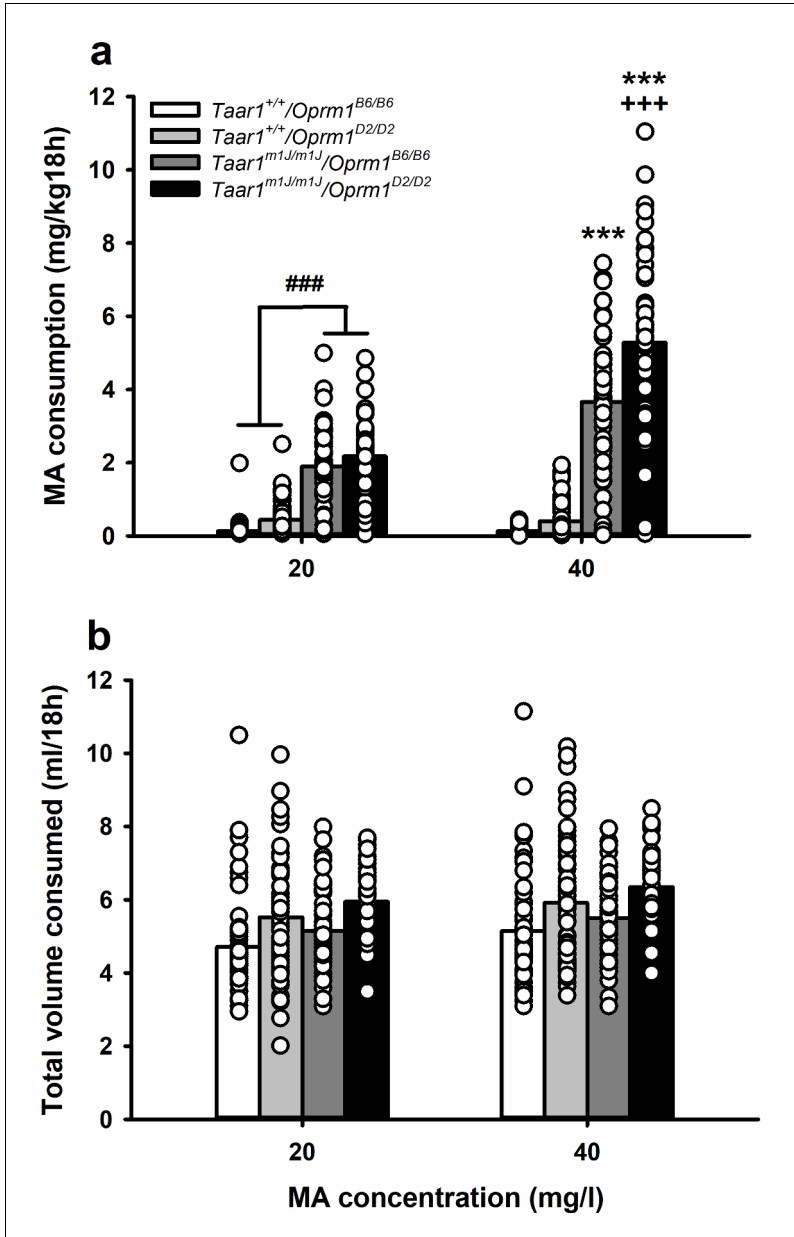

**Figure 4.** MA consumption is impacted in BXD mice by a *Taar1* x *Oprm1* interaction. (**a**) MA consumption and (**b**) total volume consumed, when MA was offered vs. water for 18h/day at a concentration of 20 or 40 mg/l for BXD strain mice with different combined *Taar1/Oprm1* genotypes. Presented are means (represented by bars) and individual data points (represented by circles) for data collapsed on sex. n = 52–71/ genotype tested in 4 cohorts of 27–97 mice. Repeated measures ANOVA, ###p<0.001 for the main effect of *Taar1*$^{+/+}$ vs. *Taar1*$^{m1J/m1J}$. Repeated measures ANOVA followed by simple main effects analysis, ***p<0.001 compared to *Taar1*$^{+/+}$ with the same *Oprm1* genotype; +++*p* < 0.001 compared to *Taar1*$^{m1J/m1J}$/*Oprm1*$^{B6/B6}$ at 40 mg/l MA. BXD, C57BL/6J (B6) x DBA2/J (D2); MA, methamphetamine; *Oprm1*, mu-opioid receptor gene; *Oprm1*$^{B6/B6}$, homozygous for B6 *Oprm1* allele; *Oprm1*$^{D2/D2}$, homozygous for D2 *Oprm1* allele; *Taar1*, trace amine-associated receptor 1 gene; *Taar1*$^{+/+}$, homozygous for reference *Taar1*$^{+}$ allele; *Taar1*$^{m1J/m1J}$, homozygous for mutant *Taar1*$^{m1J}$ allele. The raw data represented in the graphs are available in *Figure 4—source data 1*.

DOI: https://doi.org/10.7554/eLife.46472.009

The following source data is available for figure 4:

**Source data 1.** MA consumption (mg/kg/18h) and total volume consumed (ml/18h) data for male and female BXD mice, according to *Taar1* and *Oprm1* genotype.

DOI: https://doi.org/10.7554/eLife.46472.010

**Table 1.** *Taar1/Oprm1* genotype combinations for the BXD strains.

| *Taar1/Oprm1* genotype combination | MA intake | MA-induced change in body temperature |
|---|---|---|
| *Taar1$^{+/+}$/Oprm1$^{B6/B6}$* | BXD 154, 184, 196, 212, 218 | BXD 154, 184, 196, 218 |
| *Taar1$^{m1J/m1J}$/Oprm1$^{B6/B6}$* | BXD 113, 160, 171, 194 | BXD 113, 160, 171, 194 |
| *Taar1$^{+/+}$/Oprm1$^{D2/D2}$* | BXD 161, 199, 205, 215 | BXD 161, 199, 205 |
| *Taar1$^{m1J/m1J}$/Oprm1$^{D2/D2}$* | BXD 178, 186, 204, 210, 216 | BXD 178, 186, 210, 216 |

Abbreviations: BXD, C57BL/6J (B6) x DBA2/J (D2); MA, methamphetamine; *Oprm1*, mu-opioid receptor gene; *Oprm1$^{B6/B6}$*, homozygous for B6 *Oprm1* allele; *Oprm1$^{D2/D2}$*, homozygous for D2 *Oprm1* allele; *Taar1*, trace-amine associated receptor 1 gene; *Taar1$^{+/+}$*, homozygous for reference *Taar1$^{+}$* allele; *Taar1$^{m1J/m1J}$*, homozygous for mutant *Taar1$^{m1J}$* allele.

DOI: https://doi.org/10.7554/eLife.46472.011

appears also to have a primary role in the hypothermic response to MA. However, *Taar1* genotype does not impact basal body temperature, nor the hypothermic response to ethanol (*Harkness et al., 2015*). Further, the *Taar1* mutation does not appear to affect locomotor activity at baseline or in response to the 2 mg/kg dose of MA used here (*Shabani et al., 2011*; *Wheeler et al., 2009*), making it unlikely that differences in thermal response to MA in mice with different *Taar1* genotypes are due to differences in movement after treatment.

In addition to the significant QTL on chromosome 10 for MA intake and MA-induced thermal response, we identified other genomic regions at the suggestive level of significance. None of these overlapped for the two traits, and the chromosome 4 MA intake QTL that reached the suggestive significance threshold with composite mapping was not identified in our previous studies of the MA drinking lines (*Belknap et al., 2013*). There may be additional alleles that account for smaller amounts of genetic variance that our analysis did not have the power to detect. The need for large sample sizes to detect rare variants and alleles with smaller independent effects for complex traits is recognized (*Belknap, 1998*; *Belknap et al., 1996*; *Buchner and Nadeau, 2015*; *Flint, 2011*; *Solberg Woods, 2014*).

## Independent and epistatic effects of *Taar1* and *Oprm1*

Previous findings indicate that *Oprm1* regulation and opioid system activity contribute to MA consumption in mice (*Belknap et al., 2013*; *Eastwood et al., 2018*; *Eastwood and Phillips, 2014a*; *Eastwood and Phillips, 2014b*), and MA dependence/psychosis in humans (*Ide et al., 2004*; *Ide et al., 2006*). Our results in the BXD strains with different combinations of *Taar1/Oprm1* genotypes support a potential epistatic interaction in the effects of these genes on MA consumption and on the hypothermic response to MA. The combined effects of *Oprm1* and *Taar1* genotype on each trait was non-additive. *Oprm1* genotype impacted MA consumption in *Taar1$^{m1J/m1J}$* mice, but not *Taar1$^{+/+}$* mice. The opposite pattern of effect emerged for MA-induced hypothermia; *Oprm1* genotype impacted this trait in *Taar1$^{+/+}$* mice, but not *Taar1$^{m1J/m1J}$* mice. For both traits, the *Oprm1$^{D2/D2}$* mice exhibited the more robust effects, compared to *Oprm1$^{B6/B6}$* mice, and *Oprm1* genotype had an impact only in mice with the stronger MA trait. Thus, *Taar1$^{m1J/m1J}$/Oprm1$^{D2/D2}$* mice consumed more MA than *Taar1$^{m1J/m1J}$/Oprm1$^{B6/B6}$* mice, and *Taar1$^{+/+}$/Oprm1$^{D2/D2}$* mice showed greater MA-induced hypothermia compared to *Taar1$^{+/+}$/Oprm1$^{B6/B6}$* mice. Low levels or absence of the MA-related phenotype could preclude the *Oprm1* genotype from having an effect. This interaction between *Taar1* and *Oprm1* was supported by composite interval QTL mapping for MA-induced body temperature change, in which controlling for *Oprm1* genotype resulted in a significantly increased LOD score, specifically for the chromosome 10 QTL. For MA intake, the LOD score increase did not meet the significance threshold, but the strong statistical trend deserves further consideration, which could be accomplished by testing more BXD strains of appropriate genotypes, as they become available, to increase power.

When we assessed the independent associations of each genotype with these MA-related traits, significant correlations were found only with *Taar1* genotype. That *Oprm1* genotype alone did not correspond with amount of MA consumed is consistent with our previous findings, indicating that *Oprm1* genotype is not a risk factor for MA intake, and its interactive effect with *Taar1* may be

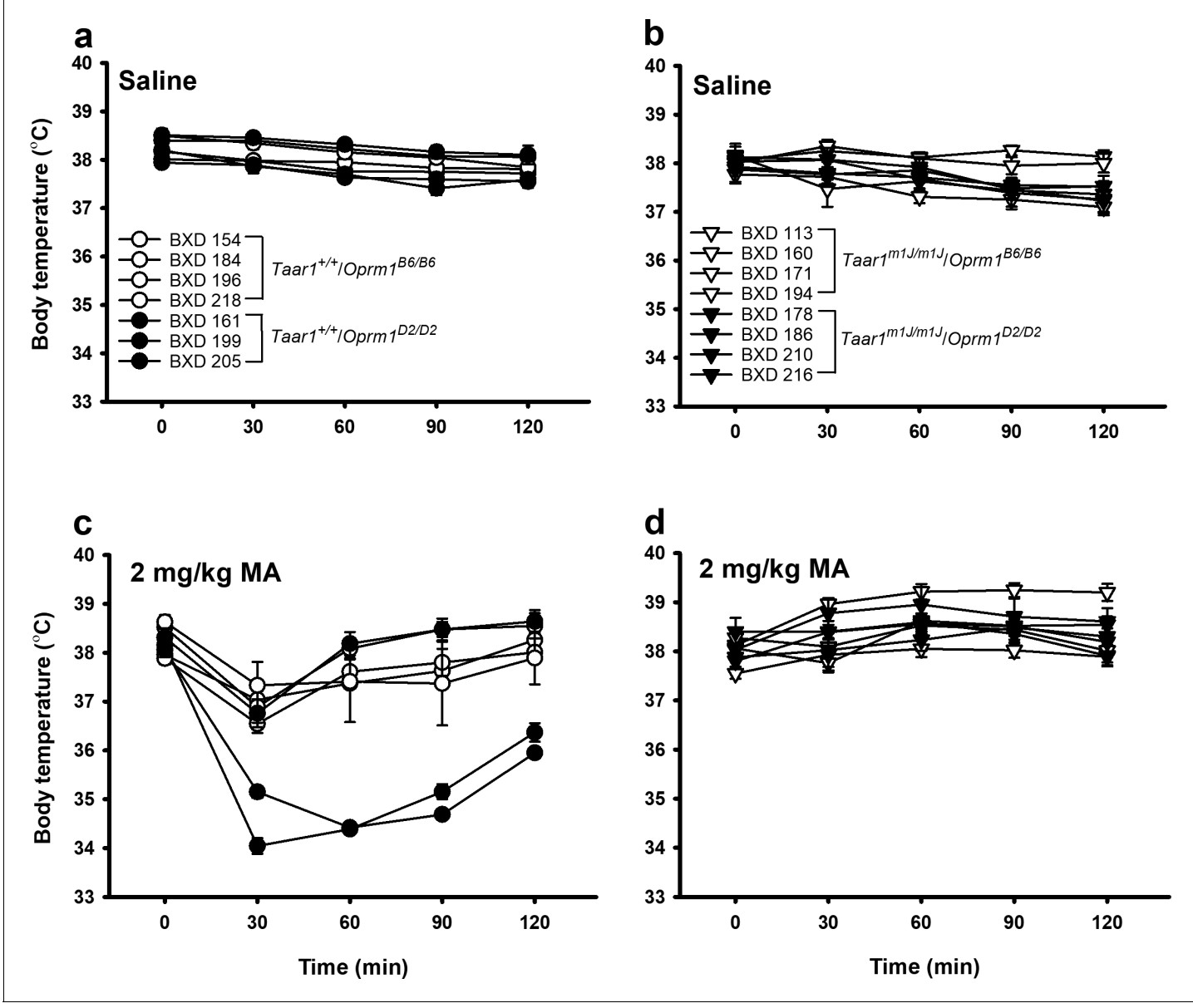

**Figure 5.** Body temperature among BXD strains treated with saline or MA is strain-dependent. Body temperatures obtained just before (T0) or 30–120 min (T30-120) after injection for (**a**) saline-treated *Taar1*$^{+/+}$ mice of either *Oprm1* genotype, (**b**) saline-treated *Taar1*$^{m1J/m1J}$ mice of either *Oprm1* genotype, (**c**) 2 mg/kg MA-treated *Taar1*$^{+/+}$ mice of either *Oprm1* genotype, and (**d**) 2 mg/kg MA-treated *Taar1*$^{m1J/m1J}$ mice of either *Oprm1* genotype. Presented are means ± SEM for data collapsed on sex. n = 4–15/strain/dose tested in 16 cohorts of 8–36 mice. For clarity, significant findings are discussed in the text. BXD, C57BL/6J (B6) x DBA2/J (D2); MA, methamphetamine; *Oprm1*, mu-opioid receptor gene; *Oprm1*$^{B6/B6}$, homozygous for B6 *Oprm1* allele; *Oprm1*$^{D2/D2}$, homozygous for D2 *Oprm1* allele; *Taar1*, trace amine-associated receptor 1 gene; *Taar1*$^{+/+}$, homozygous for reference *Taar1*$^{+}$ allele; *Taar1*$^{m1J/m1J}$, homozygous for mutant *Taar1*$^{m1J}$ allele. The raw data represented in the graphs are available in *Figure 5—source data 1*.

DOI: https://doi.org/10.7554/eLife.46472.012

The following source data is available for figure 5:

**Source data 1.** Core body temperature (℃) data across time for saline and 2 mg/kg MA-treated male and female, BXD mice, according to *Taar1* and *Oprm1* genotype.

DOI: https://doi.org/10.7554/eLife.46472.013

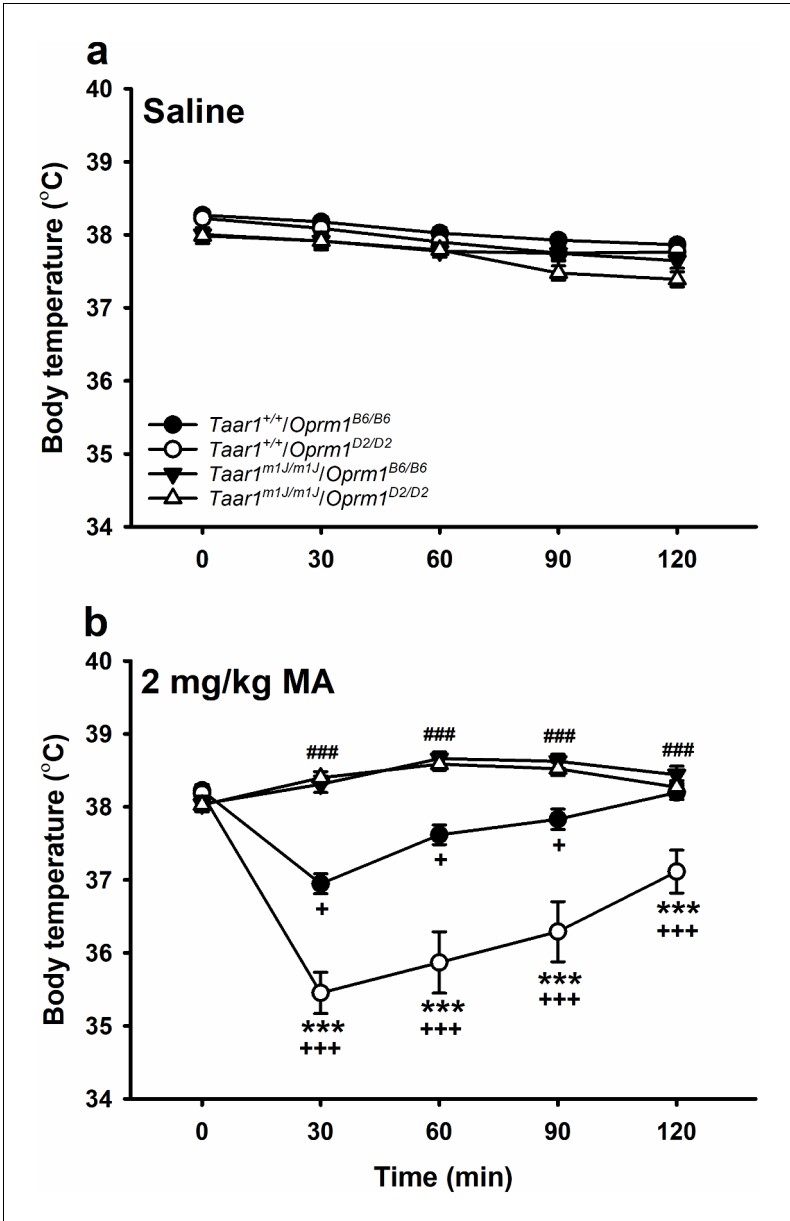

**Figure 6.** MA-induced hypothermia is impacted in BXD mice by a *Taar1* x *Oprm1* interaction. Body temperatures obtained just before (T0) or 30–120 min (T30-120) after (**a**) saline or (**b**) 2 mg/kg MA treatment for BXD strain mice with different combined *Taar1/Oprm1* genotypes. Presented are means ± SEM for data collapsed on sex. n = 30–47/genotype/MA dose tested in 16 cohorts of 8–36 mice. Repeated measures ANOVA followed by Dunnett's post hoc test,+$p < 0.05$ compared to T0 of same genotype; +++$p < 0.001$ compared to T0 of same genotype. Repeated measures ANOVA followed by simple main effects analysis, ***$p<0.001$ compared to *Taar1$^{+/+}$/Oprm1$^{B6/B6}$*. Repeated measures ANOVA collapsed on *Oprm1* genotype followed by Dunnett's post hoc test, ###$p<0.001$ compared to T0 of *Taar1$^{m1J/m1J}$* mice. BXD, C57BL/6J (B6) x DBA2/J (D2); MA, methamphetamine; *Oprm1*, mu-opioid receptor gene; *Oprm1$^{B6/B6}$*, homozygous for B6 *Oprm1* allele; *Oprm1$^{D2/D2}$*, homozygous for D2 *Oprm1* allele; *Taar1*, trace amine-associated receptor 1 gene; *Taar1$^{+/+}$*, homozygous for reference *Taar1$^{+}$* allele; *Taar1$^{m1J/m1J}$*, homozygous for mutant *Taar1$^{m1J}$* allele. The raw data represented in the graphs are available in *Figure 6—source data 1*.

DOI: https://doi.org/10.7554/eLife.46472.014

The following source data is available for figure 6:

**Source data 1.** Core body temperature (˚C) data across time for saline and 2 mg/kg MA-treated male and female, BXD mice, according to *Taar1* and *Oprm1* genotype.

DOI: https://doi.org/10.7554/eLife.46472.015

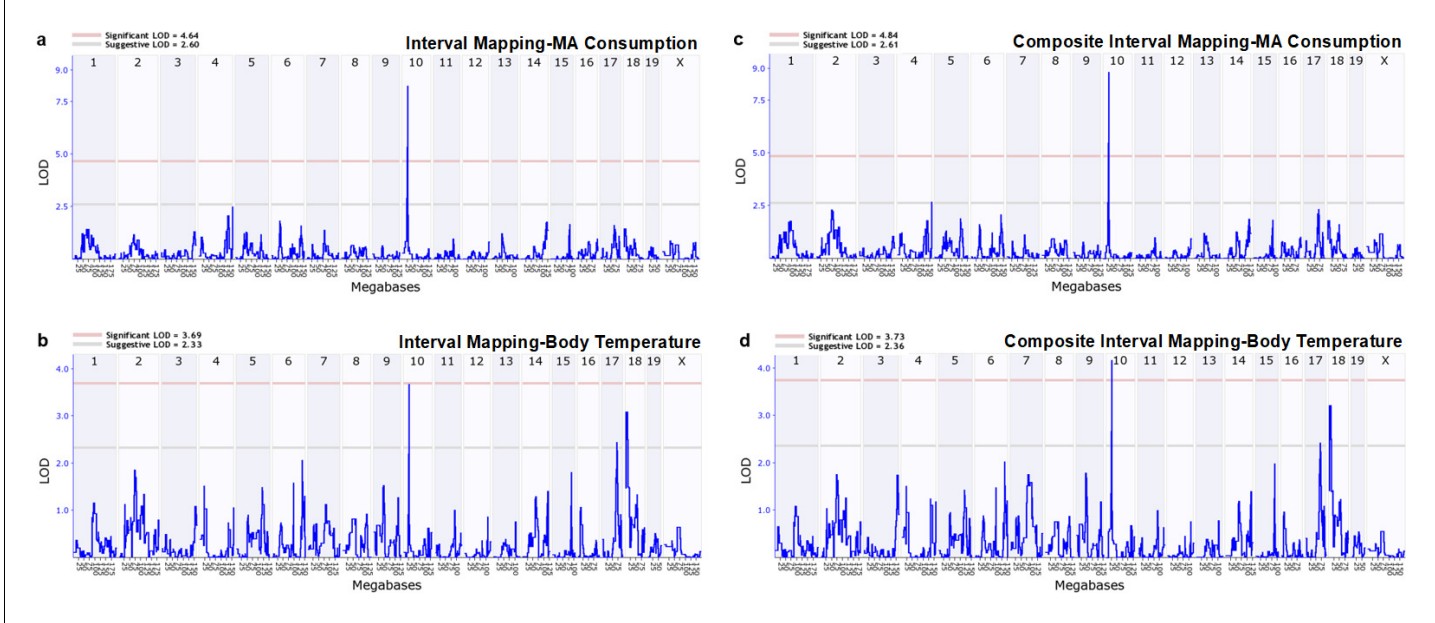

**Figure 7.** Genome-wide QTL scans identify a significant QTL on mouse chromosome 10 for both MA consumption and MA effect on body temperature. Means for 18 BXD strains were used for mapping of (**a**) consumption of MA from a 40 mg/l solution, and 15 strains for mapping of (**b**) body temperature response to 2 mg/kg MA at 30 min post-treatment. Composite interval mapping for MA consumption (**c**) and body temperature response (**d**) included *Oprm1* genotype as a co-factor. Chromosome number and megabase position are represented along the x-axis (chromosome indicated at top of plot). The y-axis represents the LOD ratio, a measure of the strength of association between variation in the phenotype and genetic differences (alleles) at a particular chromosomal locus. The horizontal pink and gray lines indicate the significant and suggestive threshold levels, respectively. Mapping results were generated using the QTL mapping module of GeneNetwork (www.genenetwork.org). BXD, C57BL/6J (B6) x DBA2/J (D2); LOD, logarithm of the odds; MA, methamphetamine; QTL, quantitative trait locus. The raw data represented in the graphs are available in *Figure 7—source data 1*.

DOI: https://doi.org/10.7554/eLife.46472.016

The following source data is available for figure 7:

**Source data 1.** BXD means for QTL mapping of MA consumption (mg/kg/18h) from the 40 mg/l concentration and for QTL mapping of 2 mg/kg MA-induced body temperature change.

DOI: https://doi.org/10.7554/eLife.46472.017

associated with the regulation of *Oprm1* by a significant MA intake risk gene network (*Belknap et al., 2013*). The interactive effect of *Oprm1* with *Taar1*, in the absence of an independent effect of *Oprm1*, is consistent with other studies demonstrating epistatic interactions involving polymorphisms that do not have significant independent associations with phenotypes (*Lehner, 2011*).

Epistasis has been proposed as a reason for variability in disease occurrence among individuals with disease risk mutations. However, like the many intermolecular epistatic interactions with unknown mechanisms (*Lehner, 2011*), the mechanism(s) underlying the impact of *Oprm1* genotype on these *Taar1*-associated traits is not yet known. One possible mechanism to explore relates to differential *Oprm1* expression. A polymorphism in the promoter region of *Oprm1* was related to promoter activity, in which D2 mice had greater *Oprm1* promoter activity compared to B6 mice (*Doyle et al., 2006*). It was hypothesized that this might alter MOP-r expression. *Doyle et al. (2014)* found greater *Oprm1* expression in the frontal cortex of D2, compared to B6 mice. It is possible that greater expression of *Oprm1* could result in greater MA intake in *Taar1^{m1J/m1J}/Oprm1^{D2/D2}* mice, and greater MA-induced hypothermia in *Taar1^{+/+}/Oprm1^{D2/D2}* mice. However, *Eastwood et al. (2018)* examined MOP-r protein density in D2 vs. B6 mice in the prefrontal cortex and did not find a difference. On the other hand, our low line mice, which consumes little MA, is sensitive to MA-induced hypothermia and is largely of the *Taar1^{+/+}* genotype, had both greater *Oprm1* expression and MOP-r density exclusively in the prefrontal cortex, compared to our high line mice, which has the opposite MA-related phenotypes and *Taar1^{m1J/m1J}* genotype (*Belknap et al., 2013*; *Eastwood et al., 2018*). Thus, the hypothesis that greater *Oprm1* expression results in greater MA-

induced hypothermia in $Taar1^{+/+}$ genotype mice is consistent with these findings, but the hypothesis that greater $Oprm1$ expression results in greater MA intake in $Taar1^{m1J/m1J}$ genotype mice is not consistent with the existing data. In fact, greater MA intake is associated with reduced prefrontal cortex $Oprm1$ and MOP-r expression in the high line, and peripheral application of drugs that increase MOP-r activity reduced MA consumption in high line mice (*Eastwood et al., 2018*; *Eastwood and Phillips, 2014a*). We are currently creating $Taar1^{+/+}$ knock-in mice on a D2 strain background and $Taar1^{m1J/m1J}$ knock-in mice on a B6 background. Epistatic behavioral and molecular effects may be more approachable for study on these homogeneous backgrounds.

## Effects of *Taar1* and *Oprm1* on total fluid intake

We observed genotype-dependent effects on total volume consumed from the MA and water tubes, but conclude that genotypic differences in MA consumption were not an artifact of overall fluid intake differences. The total volume consumption difference between $Taar1^{m1J/m1J}$ and $Taar1^{+/+}$ mice was small, compared to their robust difference for MA intake. Furthermore, the interactive effect of *Taar1* and *Oprm1* was specific to MA intake. We previously reported greater total fluid intake in high line mice, compared to low line mice, of about the same magnitude (~0.5 ml) in some, but not all, replicate sets of lines (*Hitzemann et al., 2019*; *Shabani et al., 2011*; *Shabani et al., 2019*; *Wheeler et al., 2009*). Locomotor behavior has been found to be positively associated with MA intake in the low and high lines (*Shabani et al., 2012a*), and larger intake volumes in mice consuming more MA could be related to stimulant effects that impact fluid needs. But, because a difference in total fluid intake has not always been found, another interpretation is that, due to greater reinforcing effects of MA in mice of the $Taar1^{m1J/m1J}$ genotype, such as the high line mice (*Shabani et al., 2012a*), avidity for MA is greater, leading to larger intake volumes.

## Body temperature differences unrelated to MA exposure

We observed some differences in body temperature between the knock-in and control mice that were unrelated to MA treatment. However, these baseline differences did not appear to account for the genotype-dependent MA response. Although the knock-in mice had higher baseline temperatures and therefore, a greater opportunity to experience a reduction in body temperature when treated with MA, the difference at baseline was of about 0.5℃, whereas the difference at 30 min post MA treatment was over 1℃. This was, in part, because the MA responses were qualitatively opposite, with the knock-in mice exhibiting decreases in body temperature, similar to MALDR mice, and the controls exhibiting a time-dependent increase, similar to MAHDR mice (*Harkness et al., 2015*).

## Sex differences

In our previous studies, sex differences for MA consumption and the thermal response to MA were inconsistently found, and some are summarized in *Reed et al. (2018)*. Generally, when there have been differences, females have consumed about 15% or 0.5 mg/kg more MA than males, but in no case were there differences between the MAHDR and MALDR lines or between mice with the different *Taar1* genotypes in only one sex (*Eastwood and Phillips, 2014a*; *Harkness et al., 2015*; *Reed et al., 2018*; *Shabani et al., 2011*; *Shabani et al., 2016*; *Wheeler et al., 2009*). Likewise, sex did not have a significant impact on genotype-dependent differences in MA consumption in the present studies. Similarly, there were some sex effects in the body temperature studies, but sex did not determine the pattern of response to saline vs. MA in each genotype across time. Thus, these sex effects did not impact overall interpretations of data for either trait.

## Potential limitations

One consideration regarding *Taar1* involvement in MA consumption and MA thermal response is the potential for differential potency of agonists interacting with the two TAAR1 isoforms. However, our analyses indicate that the receptor expressed by $Taar1^{m1J}$ is non-functional, rather than sub-functional. Thus, the D2-like isoform of TAAR1, expressed by $Taar1^{m1J}$, exhibits no cAMP response to a wide range of MA concentrations, or to TAAR1-specific agonists (*Harkness et al., 2015*; *Shi et al., 2016*). If the *Taar1* mutation reduces potency of agonists, we would expect a shift in the agonist dose-response curve for cAMP accumulation. Further, at doses of MA up to 16 mg/kg, high line

mice fail to exhibit MA-induced hypothermia, whereas low line mice exhibit hypothermia at doses as low as 1 mg/kg that reverses toward hyperthermia at higher doses (*Harkness et al., 2015*). Thus, the hypothermic effect of MA via TAAR1 may compete with the expression of MA-induced hyperthermia occurring via a non-TAAR1 mechanism in low mice.

The gRNA used to produce the MAHDR-*Taar1*$^{+/+}$ knock-in mice was a perfect match for the sequence on either side of the chromosome 10 *Taar1* SNP, with imperfect mapping to some additional chromosomes. In our original QTL analysis for MA drinking and in the current analysis (*Figure 7*), there were no significant or suggestive QTLs mapped to those additional chromosomes (*Belknap et al., 2013*; and data herein). This suggests that if off-target deletions or insertions occurred at these locations, they would be unlikely to have the profound impact on the MA-related traits that we have observed. The chromosome 10 QTL accounts for 60% of the genetic variance in MA intake, and *Taar1* genotype-phenotype correlations range from r = 0.81 to 0.96 for 40 mg/L MA intake, and r = 0.71 to 0.82 for 2 mg/kg MA-induced hypothermia (*Reed et al., 2018*; and data herein). Therefore, the large effects we observed would be expected for replacement of *Taar1*$^{m1J}$ with *Taar1*$^+$. The high line mice possess a genetically heterogeneous background; thus, sequencing the CRISPR knock-in and control mice would not provide specific information about off-target effects, because the existing genetic variation would not be separable from such effects. Further, there appears to be some consensus that off-target effects of CRISPR-Cas9 are rare and indistinguishable from the background rate of de novo mutation (e.g., *Anderson et al., 2018*; *Ayabe et al., 2019*; *Iyer et al., 2018*; *Mianné et al., 2016*; *Nakajima et al., 2016*; *Willi et al., 2018*).

The knock-in and control mice used in our studies were produced by independent breeding pairs. As noted in Materials and methods, this interbreeding of like-*Taar1*-genotype individuals is a consistent feature for all other existing populations in which the *Taar1* SNP exists. The only exception has been for the B6 x D2 F2 mice that were produced to create the high and low selected lines, which existed as multi-*Taar1*-genotype littermates. An equally strong association of *Taar1* genotype with MA intake was found in these mice (*Reed et al., 2018*).

## Conclusions

In summary, these data demonstrate that *Taar1* is a major contributor to MA intake and MA-induced hypothermia, and provide evidence for an interaction between *Taar1* and *Oprm1* in their effects on these MA traits. Genetic variation in the human *Oprm1* gene has been associated with MA dependence/psychosis (*Ide et al., 2004*; *Ide et al., 2006*), but additional research results in this area have not appeared in the literature. The potential impact of human *TAAR1* genetic variation on risk for MA use or on the magnitude of MA-related phenotypes is not yet known. There are over 200 nonsynonymous SNPs with the potential for missense variants in human *TAAR1* (*Rutigliano et al., 2017*) Initial examination of some of these human *TAAR1* variants indicates that TAAR1 proteins with variable levels of function are expressed (*Shi et al., 2016*). The success of TAAR1 agonists in treatment is dependent upon at least partial receptor function that can be enhanced. Whether human TAAR1 variants are relevant to MA use and addiction in humans, and whether TAAR1 agonists are a viable treatment option, are important research questions to pursue. Further, research into potential interactive effects between human *TAAR1* and *OPRM1* variants for their impact on MA use risk and response would be valuable.

## Materials and methods

### Key resources table

| Reagent type (species) or resource | Designation | Source or reference | Identifiers | Additional information |
|---|---|---|---|---|
| Strain, strain background (*Mus musculus*, females) | CD1/NCrl | Charles River | Strain code: 022 | foster dams for generation of knock-in mice |

*Continued on next page*

*Continued*

| Reagent type (species) or resource | Designation | Source or reference | Identifiers | Additional information |
|---|---|---|---|---|
| Strain, strain background (*Mus musculus*, females and males) | BXD recombinant inbred strains | University of Tennessee Health Sciences Center | | original breeders from Dr. Robert Williams; offspring for research produced by Dr. Tamara Phillips at VA Portland Health Care System (VAPORHCS) |
| Genetic reagent (*Mus musculus*, females) | MAHDR | VAPORHCS | | methamphetamine high drinking mice; created by Dr. Tamara Phillips; all are homozygous for SNP rs33645709 |
| Genetic reagent (*Mus musculus*, females and males) | MAHDR-*Taar1*$^{m1J/m1J}$ control | this paper | | created by Oregon Health & Science University Transgenic Mouse Models Shared Resource; offspring for research produced by Dr. Tamara Phillips at VAPORHCS |
| Genetic reagent (*Mus musculus*, females and males) | MAHDR-*Taar1*$^{+/+}$ knock-in | this paper | | created by Oregon Health & Science University Transgenic Mouse Models Shared Resource; offspring for research produced by Dr. Tamara Phillips at VAPORHCS |
| Biological sample (*Mus musculus*) | tail snip | other | | obtained from breeders and research animals to determine genotype |
| Sequence-based reagent | *Taar1* gRNA | this paper | | designed by ThermoFisher Scientific; reagent for generation of knock-in mice (*Mus musculus*) |
| Sequence-based reagent | 100b oligo single strand DNA for *Taar1*$^+$ (ossDNA) | this paper | MGI:MGI2148258 | synthesized by ThermoFisher Scientific; reagent for generation of knock-in mice (*Mus musculus*) |
| Sequence-based reagent | Cas9 mRNA | TriLink | Catalog number: L-7606 | reagent for generation of knock-in mice (*Mus musculus*) |
| Sequence-based reagent | primers for genotyping *Oprm1* | this paper | | original reference for these primers: *Ferraro et al. (2005)*; based on *Oprm1* sequence NM_001304955 (*Mus musculus*) |
| Sequence-based reagent | *DdeI* restriction enzyme | ThermoFisher Scientific | Catalog number: ER1882 | reagent for *Oprm1* genotyping (*Helicobacter pylori*, RFL3) |
| Commercial assay or kit | QuickExtract DNA Extraction Solution | Lucigen | Catalog number: QE09050 | |
| Commercial assay or kit | Hotstart DNA Polymerase Kit | Qiagen | Catalog number: 203205 | |

*Continued on next page*

*Continued*

| Reagent type (species) or resource | Designation | Source or reference | Identifiers | Additional information |
|---|---|---|---|---|
| Commercial assay or kit | Taqman kit for determining *Taar1* alleles | ThermoFisher Scientific | Custom order | probes based on *Taar1* sequence: NM_053205.1; SNP rs33645709 |
| Chemical compound, drug | ethidium bromide | Sigma Aldrich | Catalog number: E1510 | reagent for *Oprm1* genotyping |
| Chemical compound, drug | (+)-methamphetamine hydrochloride | Sigma Aldrich | Catalog number: M8750 | |
| Chemical compound, drug | sterile 0.9% saline | Baxter Healthcare Corporation | Catalog number: 2F7124 | vehicle for methamphetamine |
| Software, algorithm | Statistica | TIBCO Software Inc | | |
| Software, algorithm | GeneNetwork | University of Tennessee | RRID: SCR_002388 | www.genenetwork.org |
| Software, algorithm | R | The R Foundation for Statistical Computing | RRID: SCR_001905 | www.r-project.org |

## Animals

All experiments were performed in accordance with the National Institutes of Health Guidelines for the Care and Use of Laboratory Animals (*National Research Council, 2011*) and were approved by the Institutional Animal Care and Use Committee of the VA Portland Health Care System (VAPORHCS). Male and female mice were group housed (2–5 per cage) in filtered polycarbonate shoebox cages (28.5 × 17.5×12 cm) that were lined with Bed-o'Cobs bedding (The Andersons, Inc, Maumee, OH) and fitted with wire tops, except during MA drinking studies. For these studies, mice were individually housed in the same type of caging and provided with a cotton fiber nestlet for enrichment. All mice were maintained on a 12:12h light:dark schedule (lights on at 0600h), and had *ad libitum* access to laboratory rodent block food (PicoLab Laboratory Rodent Diet 5LOD, 4.5% fat content; Animal Specialties, Woodburn, OR) and tap water.

## Design of the CRISPR-Cas9 knock-in of *Taar1*$^{m1J}$ and CRISPR reagents

The *Taar1*$^{m1J}$ mouse SNP, rs33645709, encodes a non-synonymous proline to threonine mutation at amino acid position 77 that originally occurred in D2 mice in 2001–2003 as a spontaneous mutation (*Reed et al., 2018*). Proline is often highly conserved, because it is essential for conservation of a particular protein fold. In fact, this mutation renders TAAR1 non-functional, and the SNP is fixed (homozygous) in high line mice (*Harkness et al., 2015*; *Reed et al., 2018*; *Shi et al., 2016*). CRISPR-Cas9 technology (*Jinek et al., 2012*; *Zhang, 2012*) was used to replace the mutant *Taar1*$^{m1J}$ allele with the reference *Taar1*$^{+}$ allele in high line mice to generate a homozygous *Taar1*$^{+/+}$ knock-in on the high MA drinking line background at the Oregon Health & Science University Transgenic Mouse Models Shared Resource. CRISPR in vitro transcribed guide RNAs (gRNAs), targeting the specified region of *Taar1*, and donor (100b) oligo single strand DNA (ossDNA) for incorporation of the *Taar1*$^{+}$ sequence were designed and synthesized by ThermoFisher Scientific (Carlsbad, CA, USA) (*Figure 8a*). During the development of the gRNA, in silico analysis with the basic local alignment search tool or BLAST was used to assure specificity of the sequence. The gRNA chosen for use was a perfect match for *Taar1*, with no mismatches, except at the SNP location. The gRNA sequence had similarity to nine additional regions on chromosomes 1, 3, 5, 9, 11 and 19, but with 2–3 mismatches in each case. This gRNA had the sequence tctgataatgAcctgcagcatgg. The location of the SNP is indicated by the capital A, encoding the mutant genotype present in the high line. The location of the SNP within the *Taar1* ossDNA sequence, which is the reference genotype, is indicated below by the capital C, with the gRNA sequence underlined. The gene editing replaces A with C: cFF

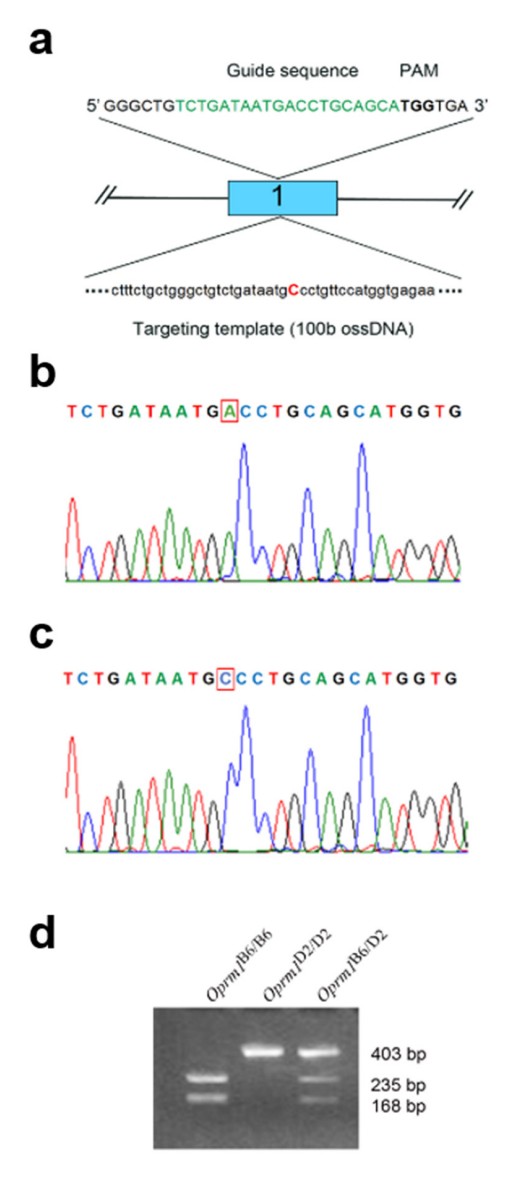

**Figure 8.** Generation of MAHDR-*Taar1*$^{+/+}$ knock-in and MAHDR-*Taar1*$^{m1J/m1J}$ control mice, and genotyping of *Taar1* and *Oprm1*. (a) Strategy for insertion of a point mutation into mouse *Taar1* exon 1. The guide sequence is indicated in green text and the protospacer adjacent motif (PAM) is indicated by bold black text. The targeting template is indicated in black text, except for the single base pair to be inserted during the CRISPR-Cas9 process, which is indicated in red text. (b) A sequencing chromatograph of a homozygous *Taar1*$^{m1J/m1J}$ mouse that is representative of all MAHDR mice. The red box indicates the single nucleotide targeted for replacement during the CRISPR-Cas9 process. (c) A sequencing chromatograph of a homozygous MAHDR-*Taar1*$^{+/+}$ edited knock-in mouse. The nucleotide that was successfully inserted during the CRISPR-Cas9 procedure is indicated by a red square. (d) An agarose genotyping gel example of *Figure 8 continued on next page*

ctggctccttcactccatggccattgtc-gactttctgctgggctgtctgataatgCcctgcagcatggtga-gaactgttgagcgctgttggtatZZt. The ossDNA was designed without a silent mutation and to be asymmetric to provide increased efficiency of the gRNA. Cas9 mRNA (catalog number L-7606) was purchased from TriLink (San Diego, CA, USA).

## Generation of MAHDR-*Taar1*$^{+/+}$ knock-in mice

High line mice were embryo donors. Three-four week old, female mice were super-ovulated, mated with high line males, and 0.5 day old eggs were isolated as previously described (*Hogan et al., 1994*). A mixture of Cas9 mRNA, *Taar1* gRNA, and donor ossDNA for *Taar1*$^{+}$ (at concentrations of 100 ng/µl, 50 ng/µl, and 100 ng/µl, respectively) was injected into the pronuclei of one-cell fertilized eggs. The injected eggs were transferred into the oviducts of pseudo pregnant recipient CD1/NCrl foster dams (Charles River, San Diego, CA, USA), and the surviving offspring were sequenced (see Genotyping and sequencing) to determine if they had retained the *Taar1*$^{m1J}$ allele or if an alteration resulting in insertion of *Taar1*$^{+}$ had occurred. Control mice (MAHDR-*Taar1*$^{m1J/m1J}$) were derived at the same time, from those mice in which the *Taar1*$^{m1J}$ allele was not successfully altered. These offspring were transported to the VAPORHCS vivarium and bred to produce the MAHDR-*Taar1*$^{+/+}$ knock-in and MAHDR-*Taar1*$^{m1J/m1J}$ control mice that participated in our experiments. Knock-in and control mice were maintained as within-line breeding pairs in a shared colony room. This breeding of individuals with the same *Taar1* genotype is consistent with the maintenance of all other populations in which the *Taar1* SNP exists (e.g., MAHDR vs. MALDR; BXD strains; D2 mice from The Jackson Laboratory vs. D2 mice from other suppliers; *Reed et al., 2018*). Representative chromatographs for the *Taar1*$^{m1J}$ (control) and *Taar1*$^{+}$ (knock-in) sequences are displayed in *Figure 8b and c*, respectively.

Separate cohorts of knock-in and control mice were tested for two-bottle choice MA intake or MA-induced change in body temperature, as detailed below and in our previous publications (*Harkness et al., 2015*; *Hitzemann et al., 2019*; *Shabani et al., 2011*; *Wheeler et al., 2009*). For MA intake, 20 knock-in and 20 control mice were tested in a single cohort (n = 10/genotype/sex), and the mice were 87–90 days old at the start of testing. For body temperature, 48 knock-in and 50 control mice were tested in 3 cohorts of 25–46 mice (final n = 12–13/genotype/dose/sex), and

*Figure 8 continued*

*Oprm1* genotypes for *Oprm1*<sup>B6/B6</sup>, *Oprm1*<sup>D2/D2</sup> and *Oprm1*<sup>B6/D2</sup> (from a B6 x D2 cross). All BXD mice were either homozygous *Oprm1*<sup>B6/B6</sup> or *Oprm1*<sup>D2/D2</sup>, as well as homozygous for one or the other *Taar1* allele. BXD, C57BL/6J (B6) x DBA2/J (D2); *Oprm1*, mu-opioid receptor gene; *Oprm1*<sup>B6/B6</sup>, homozygous for B6 *Oprm1* allele; *Oprm1*<sup>D2/D2</sup>, homozygous for D2 *Oprm1* allele; *Oprm1*<sup>B6/D2</sup>, heterozygous for B6 and D2 *Oprm1* alleles; PAM, protospacer adjacent motif; *Taar1*, trace amine-associated receptor 1 gene; *Taar1*<sup>+/+</sup>, homozygous for reference *Taar1*<sup>+</sup> allele; *Taar1*<sup>m1J/m1J</sup>, homozygous for mutant *Taar1*<sup>m1J</sup> allele.

DOI: https://doi.org/10.7554/eLife.46472.018

the mice were 79–94 days old on the day of testing.

## BXD mice

Breeding pairs that produced the BXD mice tested in these studies were obtained from RWW (University of Tennessee Health Science Center, Memphis, TN), and established within the VAPORHCS vivarium. Specific strains were chosen according to *Taar1* and *Oprm1* genotypes to allow identification of potential independent and interactive effects on MA-related phenotypes. Separate cohorts of offspring were tested for two-bottle choice MA intake or MA-induced body temperature change. Mice were homozygous for either the reference *Taar1*<sup>+</sup> or mutant *Taar1*<sup>m1J</sup> allele, as well as for the B6 or D2 *Oprm1* allele, in all possible combinations (*Table 1*). A total of 233 BXD mice (134 female and 99 male), ranging from 53 to 114 days of age, were tested for MA intake in 4 cohorts of 27–97 mice. A total of 325 BXD mice (171 female and 154 male), ranging from 49 to 124 days of age, were tested for MA-induced body temperature change in 16 cohorts of 8–36 mice. Numbers of BXD mice with the four potential combined *Taar1/Oprm1* genotypes (*Table 1*) were n = 22–45/genotype/sex for MA intake, and n = 14–25/genotype/sex/MA dose for the body temperature study.

## Genotyping and sequencing

We previously established and described genotyping methods for the *Taar1* SNP (*Harkness et al., 2015*; *Reed et al., 2018*). Genomic DNA was extracted from ear punch or tail snip samples, using QuickExtract DNA Extraction Solution (Lucigen, Middleton, WI, USA). For samples from the BXD mice and the original mice created by the CRISPR-Cas9 procedure, the *Taar1* region was PCR amplified using a Hotstart DNA polymerase kit (Qiagen, Valencia, CA, USA), with sequence-specific primers surrounding the region of interest (see *Harkness et al., 2015* for details). PCR products were sequenced at the Oregon Health & Science University Sequencing Core and aligned with the mouse *Taar1* sequence (NM_053205.1) to identify which allele(s) was present based on the rs33645709 SNP. All knock-in and control offspring of the original mice were genotyped with an updated protocol that uses a standard Taqman (ThermoFisher Scientific) with fluorescent probes to detect and differentiate the *Taar1* alleles (*Reed et al., 2018*).

We performed *Oprm1* genotyping for BXD mice as previously described (*Ferraro et al., 2005*). Sequence-specific primers for exon 10 of *Oprm1* were used as the forward primer (5'-ggttatgcctctctggattag-3') and reverse primer (5'-tccatcgcttacatcttacca-3'). A SNP in exon 10 of the B6 *Oprm1* gene (*Oprm1*<sup>B6</sup>) creates a *DdeI* restriction site (*Ferraro et al., 2005*), so that two bands (235 and 168 bp) are created when amplified DNA from mice with *Oprm1*<sup>B6</sup> is digested with the *DdeI* restriction enzyme; *DdeI* does not excise the D2 *Oprm1* gene (*Oprm1*<sup>D2</sup>), resulting in one band. After amplification, PCR products were digested with *DdeI*, run on a 4% agarose gel, and visualized using ethidium bromide staining. A representative gel is displayed in *Figure 8d* to indicate differentiation of homozygous (B6/B6, D2/D2) and heterozygous (B6/D2) *Oprm1* genotypes. However, all BXD mice used in these studies were homozygous.

## Drugs and reagents

(+)MA hydrochloride was purchased from Sigma (St. Louis, MO, USA) and dissolved in tap water for drinking or in sterile 0.9% saline (Baxter Healthcare Corp., Deerfield, IL, USA) for injection. For the body temperature studies, saline and MA were administered by intraperitoneal injection at a volume of 10 ml/kg body weight.

## Two-bottle choice MA drinking

Methods were consistent with those we used to measure two-bottle choice MA intake during the production of the MA drinking lines (*Hitzemann et al., 2019*; *Shabani et al., 2011*; *Wheeler et al., 2009*). During the initial two days of the study, mice had access to two tubes filled with tap water to familiarize them with the novel drinking apparatus. Starting on the third day, one water tube was replaced with a tube containing MA dissolved in tap water to which mice had access for 18h/day, beginning 3h before the lights turned off. The MA tube was removed for the remaining 6h of each day. Unpublished data (Phillips) indicate that the 6h forced abstinence periods between periods of 18h MA access are important for higher MA intake levels, compared to 24h MA access periods. The MA concentration was 20 mg/l for 4 days, and was increased to 40 mg/l for an additional 4 days. The MA and water tube positions were alternated every other day to account for potential side bias. MA consumption was indexed as the average intake on the second and fourth days of access to each MA concentration, as these days represent the day after the tube positions were switched, when mice should be familiar with the relative locations of the water and MA tubes. MA consumption was measured in ml (accuracy = 0.2 ml), and then converted to mg/kg, based on body weight measured every two days. Total volume consumed (ml) from the water and MA tubes during the 18h MA access periods was also measured.

## MA-induced change in body temperature

Methods for determining MA-induced change in body temperature were consistent with those we established previously (*Harkness et al., 2015*). Mice were tested after injection of saline or 2 mg/kg MA at an ambient temperature of 21 ± 2°C, for 120 min beginning at 0800h, two hours after lights on. The MA dose was chosen from previous results demonstrating that mice with functional TAAR1 exhibit a robust hypothermic response to 2 mg/kg MA 30 min after administration that is absent in mice with non-functional TAAR1 (*Harkness et al., 2015*; *Reed et al., 2018*). Treatment groups (saline or MA) were assigned so that males and females were equally represented, and mice from each family were distributed equally between the groups. Mice were weighed and then placed into individual perforated acrylic plastic cubicles (8 × 19×8 cm in W x H x L) that served to prevent huddling-associated effects on body temperature (*Crabbe et al., 1987*; *Crabbe et al., 1989*). Mice were undisturbed for 1h before a baseline rectal temperature was obtained at time 0 (T0) by inserting a glycerin-coated, 5 mm probe attached to a Thermalert model TH-8 digital thermometer (Sensortek, Clifton, NJ, USA) 2.5 cm into the rectum for 5 s. Saline or MA was then administered, and mice were returned to their cubicles. Rectal temperature was again measured at 30, 60, 90, and 120 min post-injection (T30-T120).

## QTL analysis

QTL analyses using BXD strain means were conducted using the WebQTL mapping module of GeneNetwork (www.genenetwork.org; RRID:SCR_002388). The traits mapped were: 1) MA intake when the 40 mg/l MA concentration was offered, which is the phenotype used for selective breeding (*Shabani et al., 2011*; *Wheeler et al., 2009*), and 2) MA-induced body temperature change at 30 min post-treatment. The 30 min time point is when the hypothermic effect of the 2 mg/kg dose of MA is largest (*Harkness et al., 2015*; *Reed et al., 2018*). The change score was calculated by subtracting temperatures at 30 min from baseline temperatures taken immediately before treatment. QTL mapping was initially performed using the interval mapping function and then composite interval mapping was applied, with *Oprm1* genotype at rs29351111 as a co-factor, to assess the potential interaction of *Oprm1* and *Taar1*. QTL significance was assessed using the LOD score obtained after 1000 or 2000 permutations, for interval or composite interval mapping, respectively, and was considered significant if the genome-wide *p*-value was <0.05, and considered suggestive if the genome-wide *p*-value was <0.63, which yields one false positive per genome scan on average. These are the standard settings used for GeneNetwork QTL mapping. R version 3.6.0 (The R Foundation for Statistical Computing, https://www.r-project.org/foundation/; RRID:SCR_001905) was used to analyze changes in LOD scores computed from interval vs. composite interval QTL mapping with a Chi-squared test comparing additive vs. interactive models for the effects of *Taar1* and *Oprm1* genotype. MA consumption and MA-induced body temperature change means for the BXD strains are

available in GeneNetwork. For additional information about using GeneNetwork for QTL mapping, see *Mulligan et al. (2017)*.

## Data analysis

Statistica 13.3 (TIBCO Software, Inc, Palo Alto, CA, USA) was used for statistical analyses other than QTL mapping. For MA drinking studies, MA consumption (mg/kg) and total volume consumed (ml) data were analyzed by repeated measures ANOVA, with MA concentration as the repeated measure, and genotype (or strain) and sex as possible independent variables. Body temperature data were analyzed by repeated measures ANOVA, with time as the repeated measure, and genotype (or strain), sex, and MA dose as possible independent variables. For BXD MA drinking and body temperature data, initial analyses characterizing the strains did not include sex as a factor, due to group sizes that were too small. However, analyses examining associations between *Taar1/Oprm1* genotypes and MA-related phenotypes did examine the potential impact of sex. For these analyses, to examine potential interaction effects on the measured phenotypes, data for the entire BXD population were analyzed with sex, *Taar1* genotype, and *Oprm1* genotype coded as separate independent variables (*Reed et al., 2018*). Effects were considered statistically significant at $p < 0.05$. Complex interactions were simplified by ANOVAs at levels of a particular factor. Two-way interactions were further interpreted using simple main effects analysis, with Bonferroni correction. To compare body temperature data at post-injection time points (T30-120) to T0 within a particular genotype, Dunnett's *post hoc* test was used. Pearson's r was used to calculate correlations between phenotypes and *Taar1* or *Oprm1* genotype. Sample sizes for these studies were based on our previous MA drinking and body temperature studies (e.g., *Harkness et al., 2015*; *Reed et al., 2018*).

## Acknowledgements

We thank Zhen Zhu and Nicholas Varra for assistance with data collection, and Jason Erk for assistance with data collection and mouse colony maintenance. This study was supported by NIH NIDA grants R01DA046081 (TP), P50DA018165 (TP, AJ), U01DA041579 (TP), and P30DA044223 (RW), Department of Veterans Affairs I01B × 002106 (TP), I01B × 002758 (AJ), and I01B × 003279 (KN), Oregon Health & Science University-University Shared Resources Pilot Funding to the Transgenic Mouse Models Shared Resource (KN), University of Tennessee Center for Integrative and Translational Science (RW), and the VA Research Career Scientist program (TP, AJ). The contents do not represent the views of the US Department of Veterans Affairs or the United States Government.

## Additional information

### Funding

| Funder | Grant reference number | Author |
| --- | --- | --- |
| National Institute on Drug Abuse | R01 DA046081 | Tamara J Phillips |
| National Institute on Drug Abuse | P50 DA018165 | Aaron J Janowsky<br>Tamara J Phillips |
| National Institute on Drug Abuse | U01 DA041579 | Tamara J Phillips |
| National Institute on Drug Abuse | P30 DA044223 | Robert W Williams |
| Department of Veterans Affairs | I01BX002106 | Tamara J Phillips |
| Department of Veterans Affairs | I01BX002758 | Aaron J Janowsky |
| Department of Veterans Affairs | I01BX003279 | Kim A Neve |
| Oregon Health & Science University | University Shared Resource award | Kim A Neve |
| U.S. Department of Veterans Affairs | Research Career Scientist Program - Career Scientist award | Aaron J Janowsky<br>Tamara J Phillips |

| University of Tennessee Center for Integrative and Translational Science | Center support | Robert W Williams |

The funders had no role in study design, data collection and interpretation, or the decision to submit the work for publication. The contents do not represent the views of the U.S. Department of Veterans Affairs or the United States Government.

### Author contributions
Alexandra M Stafford, Conceptualization, Data curation, Formal analysis, Investigation, Writing—original draft, Writing—review and editing; Cheryl Reed, Conceptualization, Resources, Data curation, Formal analysis, Supervision, Funding acquisition, Investigation, Methodology, Writing—original draft, Project administration, Writing—review and editing; Harue Baba, Data curation, Validation, Investigation, Visualization, Methodology; Nicole AR Walter, Conceptualization, Validation, Investigation, Visualization, Methodology, Writing—review and editing; John RK Mootz, Data curation, Formal analysis, Investigation, Visualization; Robert W Williams, Resources, Data curation, Formal analysis, Funding acquisition, Validation, Visualization, Methodology, Writing—review and editing; Kim A Neve, Conceptualization, Funding acquisition, Methodology, Project administration, Writing—review and editing; Lev M Fedorov, Conceptualization, Resources, Supervision, Visualization, Methodology, Project administration, Writing—review and editing; Aaron J Janowsky, Conceptualization, Funding acquisition, Writing—review and editing; Tamara J Phillips, Conceptualization, Resources, Formal analysis, Supervision, Funding acquisition, Validation, Investigation, Visualization, Methodology, Writing—original draft, Project administration, Writing—review and editing

### Author ORCIDs
Alexandra M Stafford (iD) https://orcid.org/0000-0003-4045-1888
Cheryl Reed (iD) https://orcid.org/0000-0002-8798-2347
Kim A Neve (iD) http://orcid.org/0000-0003-0109-7345
Tamara J Phillips (iD) https://orcid.org/0000-0002-7350-6323

### Ethics
Animal experimentation: All experiments were performed in accordance with the National Institutes of Health Guidelines for the Care and Use of Laboratory Animals and were approved by the Institutional Animal Care and Use Committee of the VA Portland Health Care System (VAPORHCS), protocol numbers 3169-14, 3169-16 and 3140-17.

### Decision letter and Author response
Decision letter https://doi.org/10.7554/eLife.46472.021
Author response https://doi.org/10.7554/eLife.46472.022

## Additional files

### Supplementary files
• Transparent reporting form
DOI: https://doi.org/10.7554/eLife.46472.019

### Data availability
All data generated or analysed during this study are included in the manuscript and supporting files. Source data files have been provided for Figures 2 through 7.

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
