## [Decision Letter]

Thank you for submitting your article "*Taar1* gene variants have a causal role in methamphetamine intake and response and interact with *Oprm1*" for consideration by *eLife*. Your article has been reviewed by three peer reviewers, and the evaluation has been overseen by a Reviewing Editor and Catherine Dulac as the Senior Editor. The following individuals involved in review of your submission have agreed to reveal their identity: Amelie Baud (Reviewer #1) and Bryan Yamamoto (Reviewer #3).

The reviewers have discussed the reviews with one another and the Reviewing Editor has drafted this decision to help you prepare a revised submission.

Summary:

The manuscript describes experiments that demonstrate the role of a DBA/2J specific *Taar1* allele on methamphetamine consumption and methamphetamine-induced hypothermia. Drawing upon previous work, the authors test the involvement of a *Taar1* allele that arose spontaneously in the DBA/2J colony at the Jackson Laboratory. Using CRISPR-Cas9 technology they generate a mouse line that replaced the DBA/2J Taar1 allele (*Taar1^m1J^*) with a wildtype allele (*Taar1^+^*) in mice selectively bred for high consumption of methamphetamine. The authors confirm the effects of *Taar1* on methamphetamine drinking as well as methamphetamine-induced hypothermia, and present evidence for an interaction between *Taar1* and a locus on chromosome 10 (*Oprm1*) using a set of BxD RI strains

Essential revisions:

1) Please use a significance threshold that takes into account the number of tests employed. This needs to be stated and the threshold justified.

2) The control line MAHDR-*Taar1^m1J^* was derived from mice in which the allele was not successfully altered during CRISPR experiments. However, it's not clear whether these lines are maintained separately or bred together – please state whether the controls are littermates of the MAHDR-*Taar1^+^* mice.

3) CRISPR can have off-target effects. The authors provide one sentence in the Materials and methods section saying there was an "initial" analysis of potential off-target sites but provide no further information. It's not clear whether any follow up studies of off-target effects were conducted. These details should be provided in the Materials and methods and Results and any potential caveats presented in the Discussion.

4) It seems a bit misleading to characterize the hypothermic effect of MA as prolonged in the *Taar1^+/+^/Oprm1^D2/D2^* mice (Figure 7B). The rate of increase in temperature AFTER the initial decrease seems fairly consistent between these mice and the *Taar1^+/+^/Oprm1^B6/B6^* mice. In comparison to T0, it may seem that the hypothermic effect is prolonged due to the significant difference in the temperature at T30, but the rate of increase wasn't measured so it's hard to conclude that Taar1*^+/+^/Oprm1^B6/B6^* mice recover more quickly.

5) Please can you explain whether the interaction of *Oprm1* genotype with *Taar1* has been examined in the QTL studies for MA drinking and hypothermia – perhaps by using *Oprm1* genotype as a covariate? Although there may not have been enough power to study gene x gene interactions due to sample size, it would be good to see this effect confirmed (or if not explain if this is due to low power)

6) The discussion about the relationship between MA consumption and body temperature (section “Relationship Between MA consumption and Body Temperature”) is confusing. It makes sense that MA-induced hypothermia could result in reduced MA drinking due to aversion, but the alternative hypothesis that MA drinking somehow alters MA-induced hypothermia is not so obvious. More explanation would be beneficial as to how *Taar1* mutation might alter meth consumption. For example if these mutants cannot appropriately thermoregulate, do they have a depressed phenotype, or have an altered mesocorticolimbic pathway?

7) The authors make several comments in the discussion about the potential translational importance of *Taar1* in human MA use and as a potential therapeutic. They should include a few sentences in the discussion about MA drinking in rodents as a model for human MA use. Are the pharmacokinetics of MA following oral consumption similar to normal human routes of MA administration?

8) One consideration that was not addressed was whether the genetic modifications were simply affecting the potency of METH. If METH acts on *Taar1*, then it seems reasonable that a non-functional *Taar1* would require higher METH intake to produce a similar effect. Please comment on this suggestion.

---

## [Author Response]

Essential revisions:1) Please use a significance threshold that takes into account the number of tests employed. This needs to be stated and the threshold justified.

We have now used a Bonferroni correction to account for multiple tests in the revised manuscript, and have stated this in the “Data analysis” section; p-values have been adjusted throughout. These corrections for multiple tests did not change our conclusions. The only changes from the initial submission involved the saline group in the BXD hypothermia study, and the *Taar1^m1J/m1J^*mice, which do not exhibit MA-induced hypothermia. Thus, none of the results for MA effects were impacted.

2) The control line MAHDR-Taar1m1J was derived from mice in which the allele was not successfully altered during CRISPR experiments. However, it's not clear whether these lines are maintained separately or bred together – please state whether the controls are littermates of the MAHDR-Taar1+ mice.

The CRISPR knock-in and control lines were maintained as within-line breeding pairs in a shared colony room. This breeding of individuals with the same *Taar1* genotype is consistent with the maintenance of all other populations in which the *Taar1* SNP exists (e.g., MAHDR vs. MALDR; BXD strains; D2 mice from The Jackson Laboratory vs. D2 mice from other suppliers; Reed et al., 2018). We now state this in the section “Generation of MAHDR-*Taar1^+/+^* knock-in mice”. Please note that when an F2 was created between the DBA/2J and C57BL/6J mice, resulting in littermates with different *Taar1* genotypes, an equally strong association of *Taar1* genotype with MA intake was found (see Reed et al., 2018). We have added a discussion point in the section “Potential Limitations”.

3) CRISPR can have off-target effects. The authors provide one sentence in the Materials and methods section saying there was an "initial" analysis of potential off-target sites but provide no further information. It's not clear whether any follow up studies of off-target effects were conducted. These details should be provided in the Materials and methods and Results and any potential caveats presented in the Discussion.

We have now included more detailed methodological information in the paper. We have also included discussion regarding potential off-target effects. In our situation, in which the sequence change was created on a heterogeneous background, sequencing the rest of the genome of the CRISPR knock-in and control mice would not provide specific information about off-target effects. This is because the existing genetic variance would not be separable from CRISPR-induced effects. A study published in Nature Methods (Schaefer et al., 2017) reported extensive off-target effects of CRISPR and has since been retracted, due to significant concerns raised in multiple commentaries *(Kim et al., 2018; Lareau et al., 2018; Lescarbeau et al., 2018; Nutter et al., 2018; Wilson et al., 2018). There appears to be some consensus that off-target effects are rare and indistinguishable from the background rate of de novo mutation (Anderson et al., 2018; Ayabe et al., 2018; Iyer et al., 2018; Mianne et al., 2016; Nakajima et al., 2016; Willi et al., 2018). We have mentioned this consensus in the paper. In addition, we describe our use of the basic local alignment search tool (BLAST) to examine potential sequence identity for the guide RNAs generated by ThermoFisher and give details about the one that we chose that was a perfect match, except for the relevant SNP. The gRNA sequence had similarity to 9 additional regions on chromosomes 1, 3, 5, 9, 11 and 19, but with 2-3 mismatches in each case (see t Design of the CRISPR-Cas9 knock-in of *Taar1^m1J^* and CRISPR reagents”). As can be seen in Figure 8 of the current paper, and in our published work (Belknap et al., 2013), there are no mapped QTLs for MA drinking or MA-induced hypothermia on any of these chromosomes. Therefore, even if there was a genetic change in any of these locations, it would be extremely unlikely that it would produce the large changes in both MA intake and MA-induced hypothermia that we observed. On the other hand, because the chromosome 10 QTL accounts for 60% of the genetic variance in MA intake and *Taar1* genotype-phenotype correlations range from r=0.81 to 0.96 for 40 mg/L MA intake and r=0.71 to 0.82 for 2 mg/kg MA-induced hypothermia (Reed et al., 2018; and data herein), the large effects we observed would be expected for replacement of *Taar1^m1J^* with *Taar1^+^*. Please see this added discussion in the section “Potential Limitations”.

*Schaefer and Commentary citations:

Kim ST, Park J, Kim D, Kim K, Bae S, Schlesner M, Kim JS. 2018. Response to “Unexpected mutations after CRISPR-Cas9 editing in vivo.” Nat Methods 15: 239-240. doi:10.1038/nmeth.4554

Lareau CA, Clement K, Hsu JY, Pattanayak V, Joung JK, Aryee MJ, Pinello L. 2018. Response to “Unexpected mutations after CRISPR-Cas9 editing in vivo.” Nat Methods 15: 238-239. doi:10.1038/nmeth.4541

Lescarbeau RM, Murray B, Barnes TM, Bermingham N. 2018. Response to “Unexpected mutations after CRISPR-Cas9 editing in vivo.” Nat Methods 15: 237. doi:10.1038/nmeth.4553

Nutter LMJ, Heaney JD, Lloyd KCK, Murray SA, Seavitt JR, Skarnes WC, Teboul L, Brown SDM, Moore M. 2018. Response to “Unexpected mutations after CRISPR-Cas9 editing in vivo.” Nat Methods 15: 235-236. doi:10.1038/nmeth.4559

Wilson CJ, Fennell T, Bothmer A, Maeder ML, Reyon D, Cotta-Ramusino C, Fernandez CA, Marco E, Barrera LA, Jayaram H, Albright CF, Cox GF, Church GM, Myer VE. 2018. Response to “Unexpected mutations after CRISPR-Cas9 editing in vivo.” Nat Methods 15:236-237. doi:10.1038/nmeth.4552

Schaefer KA, Wu WH, Colgan DF, Tsang SH, Bassuk AG, Mahajan VB. 2017. Unexpected mutations after CRISPR-Cas9 editing in vivo. Nat Methods 14:547-548. doi:10.1038/nmeth.4293

4) It seems a bit misleading to characterize the hypothermic effect of MA as prolonged in the Taar1^+/+^/Oprm1^D2/D2^ mice (Figure 7B). The rate of increase in temperature after the initial decrease seems fairly consistent between these mice and the Taar1^+/+^/Oprm1^B6/B6^ mice. In comparison to T0, it may seem that the hypothermic effect is prolonged due to the significant difference in the temperature at T30, but the rate of increase wasn't measured so it's hard to conclude that Taar1^+/+^/Oprm1^B6/B6^ mice recover more quickly.

We see the reviewers’ point and agree. We have edited the document regarding this point in the section “MA-induced Hypothermia is Impacted in BXD Mice by a *Taar1* x *Oprm1* Interaction”, and now only comment on the difference in the magnitude of effect between *Taar1^+/+^/Oprm1^D2/D2^*and *Taar1^+/+^/Oprm1^B6/B6^*mice, as this was a significant result. We note that the rate of return to baseline is similar between the two genotypes.

5) Please can you explain whether the interaction of Oprm1 genotype with Taar1 has been examined in the QTL studies for MA drinking and hypothermia – perhaps by using Oprm1 genotype as a covariate? Although there may not have been enough power to study gene x gene interactions due to sample size, it would be good to see this effect confirmed (or if not explain if this is due to low power)

Thank you for making this point! We now include composite interval QTL maps (Figure 8C-D) for MA consumption and hypothermia that used *Oprm1* genotype as a co-factor. This resulted in increased LOD scores for both traits, that was significant for the thermal trait and a strong trend for the MA consumption traits. This additional analysis strengthens our conclusions that formerly were based on the behavioral results. We now describe the composite mapping in the Materials and methods (section “QTL Analysis”), Results (section”MA Consumption and Body Temperature Response to MA Map to a Region of Chromosome 10 at the Location of *Taar1*”), and Discussion (section “Independent and Epistatic Effects of *Taar1* and *Oprm1*”).

6) The discussion about the relationship between MA consumption and body temperature (section “Relationship Between MA consumption and Body Temperature”) is confusing. It makes sense that MA-induced hypothermia could result in reduced MA drinking due to aversion, but the alternative hypothesis that MA drinking somehow alters MA-induced hypothermia is not so obvious. More explanation would be beneficial as to how Taar1 mutation might alter meth consumption. For example if these mutants cannot appropriately thermoregulate, do they have a depressed phenotype, or have an altered mesocorticolimbic pathway?

It is unlikely that the difference in MA effect on body temperature is due to inability to thermoregulate for several reasons. Mice with the *Taar1^m1J/m1J^*genotype do not differ in baseline temperatures or changes in those temperatures over time, compared to mice with the *Taar1^+/+^*genotype (Harkness et al., 2015, and data in the current manuscript). In addition, mice of the 2 genotypes exhibit comparable time-dependent patterns of response to other drugs with hypothermic effects, including ethanol (Harkness et al., 2015) and cocaine (unpublished data). Thus, the mutants display a hypothermic response, and the lack of hypothermia in response to MA is a MA-specific effect. Further, the *Taar1* mutation does not appear to affect locomotor activity at baseline or in response to this dose of MA (Shabani et al., 2011; Wheeler et al., 2009), so it is not likely that *Taar1* effects are due to differences in movement after treatment. We now discuss this in the section “Relationship Between MA Consumption and Body Temperature”. Also, we discuss other possible mechanisms by which TAAR1 may affect MA consumption (including alterations in the mesocorticolimbic pathway) in the last paragraph of the section “Taar1 and MA Consumption”.

We have not yet examined body temperature changes occurring during the time that MA is being consumed. We have observed that low line mice consume similar amounts of MA as high line mice on the first day that MA is offered, but then decrease their consumption on the subsequent day, remaining at low intake levels from then on (Shabani et al., 2012a; Eastwood et al., 2014). It is possible that once MA reaches a high enough dose during the first drinking session, that the low line mice experience hypothermia and that this plays a role in their reduced intake on subsequent days. We are in the process of obtaining equipment to allow us to examine body temperature simultaneously with locomotor activity during drinking sessions, which will allow us to consider these questions. We now discuss this in the section “Relationship Between MA Consumption and Body Temperature”.

7) The authors make several comments in the Discussion about the potential translational importance of Taar1 in human MA use and as a potential therapeutic. They should include a few sentences in the Discussion about MA drinking in rodents as a model for human MA use. Are the pharmacokinetics of MA following oral consumption similar to normal human routes of MA administration?

We discuss this in Shabani et al. (2012a), but agree that it is important to reiterate. Please see our discussion of the relevance of this mouse model in the section “*Taar1* and MA Consumption”.

8) One consideration that was not addressed was whether the genetic modifications were simply affecting the potency of METH. If METH acts on Taar1, then it seems reasonable that a non-functional Taar1 would require higher METH intake to produce a similar effect. Please comment on this suggestion.

If the mutation produced a sub-functional receptor, this would of course be possible. However, it appears that the receptor produced is non-functional. The D2-like isoform of TAAR1, expressed by *Taar1^m1J^*, exhibits no cAMP response to a wide range of MA concentrations or to other TAAR1 agonists (Harkness et al., 2015; Shi et al., 2016). We have shown that, even at high doses of MA (up to 16 mg/kg), high line mice fail to exhibit hypothermia, whereas MALDR mice express hypothermia at doses as low as 1 mg/kg (Harkness et al., 2015). Thus, we are fairly certain that the genetic modification did not simply affect potency. We now discuss this in the section “Potential Limitations”.